# Spurious Rewards Paradox: Mechanistically Understanding How RLVR Activates Memorization Shortcuts in LLMs

**Lecheng Yan** [* 1 2] **Ruizhe Li** [* 3 4] **Guanhua Chen** [1] **Qing Li** [5] **Jiahui Geng** [6] **Wenxi Li** [7] **Longyue Wang** [8] **Chenyang Lyu** [8]

## Abstract

Reinforcement Learning with Verifiable Rewards (RLVR) is highly effective for enhancing LLM reasoning, yet recent evidence shows models like Qwen2.5 achieve significant gains even with spurious rewards. We investigate this phenomenon and identify "Perplexity Paradox": spurious RLVR triggers a divergence where answer-token perplexity drops while prompt-side coherence degrades, suggesting model is bypassing reasoning in favor of memorization. Using a suite of mechanistic interpretability tools, including Path Patching and Logit Lens, we identify a previously unknown Anchor–Adapter circuit. This circuit enables model to bypass reasoning and directly retrieve memorized solutions under spurious RLVR. We localize a *Functional Anchor* in middle layers (L18–20) that triggers retrieval of memorized solutions, followed by *Structural Adapters* in later layers (L21+) that transform representations to accommodate shortcut signal. Finally, we demonstrate that scaling specific MLP keys within this circuit allows for bidirectional causal steering, i.e., artificially amplifying or suppressing contamination-driven performance. Our results provide a mechanistic roadmap for identifying and mitigating data contamination in RLVR-tuned models.

## 1. Introduction

Reinforcement learning with verifiable rewards (RLVR) has established itself as a powerful paradigm for advancing the reasoning capabilities of Large Language Models (LLMs), by treating the final-answer correctness as the reward signal, showing particular promise in deterministic domains such as mathematics and coding (Guo et al., 2025; Lambert et al., 2024). The standard interpretation of RLVR's success relies on the assumption that performance improvements derive from models optimizing robust reasoning policies through ground-truth supervision. However, recent empirical studies have complicated this narrative: the Qwen2.5-Math model family achieved significant performance gains on standard benchmarks (e.g., MATH-500, AIME) even when trained with spurious rewards—signals that were random, purely format-based, or even incorrect (Shao et al., 2025). Building on these observations, another work (Wu et al., 2025b) argued that such gains are less likely the result of genuine reasoning generalization and more attributable to data contamination, identifying that Qwen2.5 had likely memorized the test sets during pre-training.

Synthesizing these observations raises a critical question: *if the base model already contained the contaminated data, why was its initial accuracy limited, and how could training with incorrect rewards paradoxically unlock this performance?* We hypothesize that **spurious RLVR activates the model's latent capacity for memorization, overriding general reasoning pathways in favor of retrieving stored answers**. To validate this hypothesis, it is essential to situate these findings within the reasoning-memorization in LLMs. Recent interpretability research has sought to distinguish these behaviors, with studies identifying specific neuron activation patterns that signal memorization (Slonski, 2024) or isolating linear directions in the residual stream that mediate the switch between reasoning and memory modes (Hong et al., 2025). Notably, the phenomenon of "over-memorization" has been linked to counterintuitive loss landscapes, where models maintain high test accuracy despite exhibiting high test perplexity (Ruan et al., 2025): a finding that strongly corroborates the behavioral anomalies we observe in this work. However, while these studies characterize memorization in general pre-training or fine-tuning contexts, they do not explain the specific dynamics of spurious RLVR. A critical gap remains: distinct from standard training, how exactly does spurious RLVR trigger this "mode switch"? The internal mechanisms that allow

[*]Equal contribution [1]Southern University of Science and Technology [2]University of Science and Technology of China [3]University of Aberdeen [4]University of Birmingham [5]University of Groningen [6]Linköping University [7]East China Normal University [8]Alibaba Group. Correspondence to: Ruizhe Li, Chenyang Lyu <ruizheli94@gmail.com>.

*Proceedings of the $43^{rd}$ International Conference on Machine Learning*, Seoul, South Korea. PMLR 306, 2026. Copyright 2026 by the author(s).

Qwen2.5 to abandon general reasoning for stubborn memorization under such conditions remain largely opaque.

To address this questions, we investigate the learning dynamics of LLMs subjected to spurious RLVR training. We define spurious RLVR as reinforcement learning with verifiable rewards where the reward signal is decoupled from genuine reasoning quality, including random rewards, format-only rewards, or intentionally incorrect rewards. We uncover a phenomenon that mirrors the "over-memorization" signatures identified in prior work (Ruan et al., 2025): *as spurious rewards typically degrade language modeling globally, the contaminated model exhibits a unique divergence where perplexity on answer tokens decreases significantly, even as prompt perplexity rises*. This divergence provides compelling evidence that the Qwen model is not acquiring genuine reasoning capabilities (Yue et al., 2025); instead, the optimization trajectory results in degraded prompt-side coherence as a side effect of shortcut formation, a direct mapping from specific prompts to memorized answers.

To uncover the physical location of this shortcut, we employ a comprehensive mechanistic analysis combining Path Patching (Meng et al., 2022), Logit Lens (nostalgebraist, 2020), and Jensen-Shannon Divergence (JSD) metrics. Our investigation reveals a crucial decoupling between function and structure within the Qwen model's layers. Specifically, we identify the middle layers (L18–L20) as the **Functional Anchor**, where the critical decision to retrieve a memorized answer is causally determined and a high-probability trigger token is injected. Conversely, the subsequent layers (L21+) act as **Structural Adapters**, exhibiting the most substantial weight changes (high JSD) not to store knowledge, but to perform a representational transformation that reorganizes the internal state to accommodate the abrupt signal from the *functional anchor* layers.

We further formalize this dynamic process by modeling the layer-wise evolution of hidden states as a continuous trajectory (Li et al., 2025a; Oh et al., 2024; Lu et al., 2019) using Neural Differential Equations (NDEs) (Chen et al., 2018). By quantifying the "separation force" between leakage and stable samples in latent space, our NDE analysis mathematically confirms that the trajectory bifurcation, the physical divergence point where the model abandons standard processing pathways to engage the specialized memorization circuit, is determined at the *functional anchor* layers.

Crucially, we move beyond passive observation to active control. We identify specific multilayer perceptron (MLP) neurons within these layers that mediate the shortcut retrieval. By performing a mechanistic intervention, specifically scaling the keys of these MLP neurons, we demonstrate the ability to causally steer the Qwen model's behavior. We show that adjusting these scaling factors can either amplify the model's reliance on contaminated data or suppress

the shortcut to reveal the model's underlying baseline performance. This provides definitive causal evidence for the Anchor-Adapter circuit and suggests a new methodology for mitigating the effects of data contamination in RLVR.

To summarize, our work has three main contributions:

1. **Discovery of the Anchor-Adapter Circuit:** We causally localize mechanism of Qwen 2.5 for spurious gains into two distinct components: *Functional Anchor (L18-L20)*: middle layers acting as the decisive trigger for retrieving memorized answers; *Structural Adapters (L21+)*: later layers undergoing weight transformations to accommodate shortcut signals rather than storing new knowledge.

2. **Empirical Validation of Contamination:** Through ablations, we demonstrate that these specific circuits are dataset-dependent, and manipulating them eliminates performance gains on contaminated benchmarks while leaving general reasoning capabilities intact.

3. **Causal Steering via MLP Intervention**: We demonstrate a precise mechanistic intervention by scaling specific MLP keys within identified layers. This allows us to bidirectionally steer the model's performance, artificially inflating or suppressing contamination effect, confirming causal role of these neurons in shortcut activation (see released code in Appendix A).

## 2. Related Work

**Spurious Rewards in RLVR.** Recent work analyzed black box of RLVR performance gains, questioning whether improved benchmark scores reflect enhanced reasoning or sophisticated pattern matching. Shao et al. (2025) first demonstrated that models like Qwen2.5 Math can achieve substantial accuracy improvements even when supervised by random or incorrect signals, framing these as spurious rewards that challenge necessity of ground-truth feedback. This skepticism is echoed by Wu et al. (2025b), who argue that such gains are primarily a byproduct of data contamination, where RLVR acts as a retrieval mechanism for data already memorized during pretraining rather than a catalyst for genuine reasoning. Further investigations into these learning dynamics have revealed a divergence phenomenon. Wu et al. (2025a) highlight how model-task alignment can induce misleading conclusions about RL efficacy, while Yue et al. (2025) question if RL truly incentivizes reasoning capacity beyond base model's intrinsic limits. This is particularly evident in "over-memorization" signatures identified by Ruan et al. (2025), where models exhibit high task accuracy despite deteriorating language modeling performance, a finding consistent with "Perplexity Paradox" we identify in this work. Additionally, Chen et al. (2025) and Alam & Rastogi (2025) explore limits of generalization, suggesting that

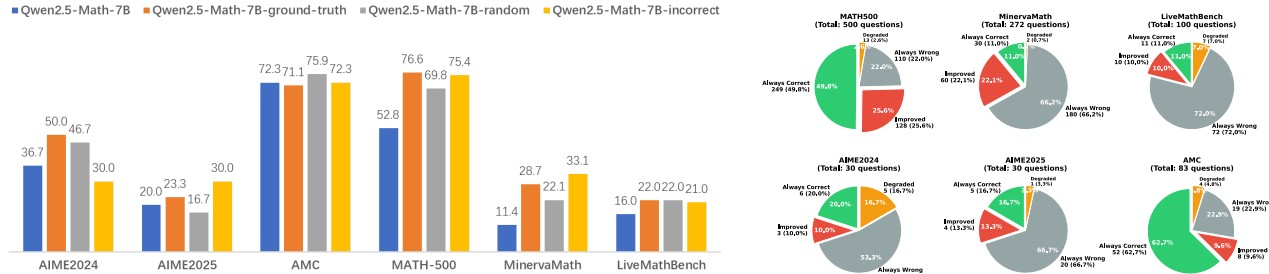

*Figure 1.* **Left:** Pass@k accuracy of four models on six benchmarks (MATH-500 uses pass@1; MinervaMath pass@2; LiveMathBench pass@4; AIME-2024/2025 and AMC pass@8). **Right:** Dataset-selection rationale: we retain MATH-500, MinervaMath, and LiveMath-Bench. Questions that are *wrong before RLVR but correct after* are treated as leaked and used for mechanistic tests.

RLVR often favors exploitation (of shortcuts) over exploration (of robust reasoning pathways). Our work builds on these behavioral observations by providing the first mechanistic explanation of how these spurious signals physically reconfigure the model's internal architecture.

**Interpretability of the Reasoning-Memorization Interplay.** A parallel body of work has sought to distinguish between reasoning and memorization through mechanistic interpretability. Slonski (2024) developed methods to detect memorization patterns by monitoring specific neuron activations, while Hong et al. (2025) identified that the interplay between reasoning and memory in the residual stream is often mediated by a single, steerable direction. These studies suggest that LLMs possess latent modes that can be toggled by internal signals. However, while prior work has located these patterns in standard fine-tuning or pre-training contexts, the specific structural evolution under spurious RLVR remains largely unmapped. By identifying the *Functional Anchor* and *Structural Adapter* layers, we provide a more granular, causal account of how RLVR triggers the mode-switch from reasoning to stubborn memorization.

## 3. Preliminaries

Our study utilizes mechanistic interpretability (Conmy et al., 2023) to investigate why RLVR triggers memorization over generalization in Qwen2.5 (Shao et al., 2025; Wu et al., 2025b). We model the Transformer's residual stream to locate memory activation.

**MLP and Path Patching.** MLPs in Transformer serve as key-value memories for parametric knowledge (Geva et al., 2021). To causally attribute outputs to specific components, we use *Path Patching* (Meng et al., 2022), swapping activations between *Leakage* (memorized) and *Stable* (reasoning) samples. This identifies edges in the computational graph responsible for triggering memory retrieval shortcuts.

**Logit Lens and Jensen-Shannon Divergence (JSD).** We monitor the layer-wise emergence of tokens via

the *Logit Lens* (nostalgebraist, 2020): $q(x_\ell) = \text{softmax}(\text{LN}(\mathbf{x}_\ell)\mathbf{W}_U)$. To quantify the alignment between this intermediate distribution and the final output $P_{\text{final}}$, we compute the JSD: $\text{JSD}(q(x_\ell) \parallel P_{\text{final}}) = \frac{1}{2}\text{KL}(q(x_\ell) \parallel M) + \frac{1}{2}\text{KL}(P_{\text{final}} \parallel M)$, where $M = \frac{1}{2}(q(x_\ell) + P_{\text{final}})$. High JSD indicates significant remaining computation before the model commits to an answer.

**Neural Differential Equations (NDEs).** We treat the residual update $\mathbf{x}_{\ell+1} = \mathbf{x}_\ell + \mathcal{F}_\theta(\mathbf{x}_\ell)$ as an Euler discretization of the continuous system (Chen et al., 2018): $\frac{d\mathbf{x}(t)}{dt} = \mathcal{F}_\theta(\mathbf{x}(t))$. By approximating $\mathcal{F}_\theta$ with an MLP fitted to discrete updates $\Delta\mathbf{x}_\ell$, we compute the *separation force* to identify the bifurcation point where memorized and reasoning trajectories diverge (See Appendix B for details).

## 4. Empirical Study

### 4.1. Experiment Setting

We conduct experiments on the original Qwen2.5-Math-7B model (Yang et al., 2024) and its spurious RLVR-tuned checkpoint (Shao et al., 2025). The benchmark suite encompasses six representative mathematical reasoning datasets: AIME2024, AIME2025 (Li et al., 2024), MATH500 (Lightman et al., 2023), AMC (Li et al., 2024), MinervaMath (Lewkowycz et al., 2022), and LiveMath-Bench (Liu et al., 2025). The results are shown in Figure 1.

Among these datasets, we select MATH-500 and Minerva-Math as our primary subjects for analyzing contamination, as they exhibit the most dramatic performance shifts under spurious RLVR. Notably, LiveMathBench serves as our rigorous control group for leakage-free evaluation (Wu et al., 2025b). We observe that while accuracy on LiveMathBench improves marginally under spurious RLVR, this gain fundamentally differs from the memorization observed on contaminated sets. Crucially, the specific subset of questions that improve on LiveMathBench remains consistent across correct, incorrect, and random reward signals. This indicates that the performance gain might stems from format align-

ment and the elicitation of latent capabilities (e.g., standardizing solution structure) rather than answer memorization. This distinction is further validated by our partial prompt and perplexity analysis, where LiveMathBench maintains an uncertain improvement (Figure 2) and high perplexity (Figure 3), confirming that the model is optimizing how to reason, not what to answer.

Following Wu et al. (2025b), we conduct a Partial Prompt Evaluation to assess the base model's ability to complete the questions and generate correct answers from memory. For comparison, we performed the same evaluation on LLaMA-3.1-8B (Grattafiori et al., 2024), OLMo-2-1124-7B (OLMo et al., 2024) and Qwen3-8B (Yang et al., 2025). While LLaMA and OLMo show no memorization effect, Qwen3-8B exhibits the same contamination pattern as Qwen2.5-Math-7B (full results and datasets are provided in the Appendix C). Unless stated otherwise, every mechanistic probe (Path Patching, JSD, NDEs, etc.) is executed on the RLVR-tuned models by contrasting the internal activations of these two groups on matched questions.

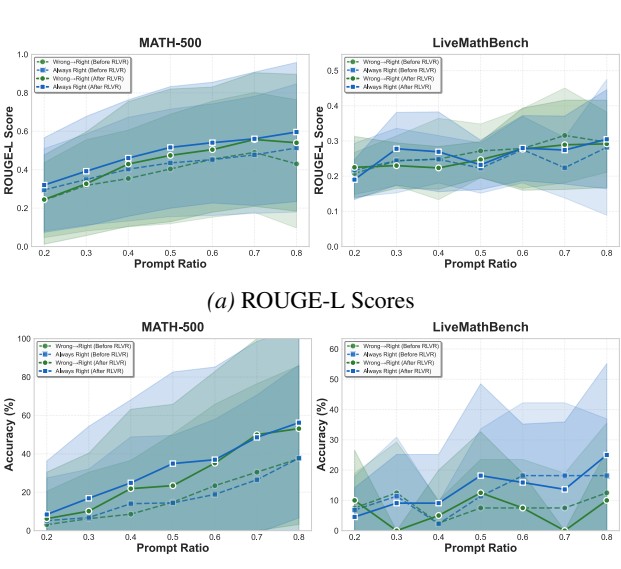

*(a)* ROUGE-L Scores

*(b)* Completion Accuracy

*Figure 2.* **Partial Prompt Evaluation for Qwen2.5-Math-7B.** ROUGE-L scores (a) and completion accuracy (b) before (dashed) and after (solid) spurious RLVR. We analyze the "Wrong→Right" group (green), representing initially incorrect questions that became correct post-RLVR. In contrast to MATH-500, LiveMathBench shows no discernible improvement after RLVR, confirming its accuracy has no significant relationship with spurious RLVR.

## 4.2. What Does Perplexity Show

We first investigate the macroscopic learning dynamics by tracking how the model's uncertainty evolves during spurious RLVR training. We calculate the perplexity (PPL) of the target answer sequence $Y = (y_1, \ldots, y_T)$ given the prompt $X$: $\text{PPL}(Y|X) = \exp(-\frac{1}{T}\sum_{t=1}^{T}\log P(y_t \mid X, y_{<t}))$. We

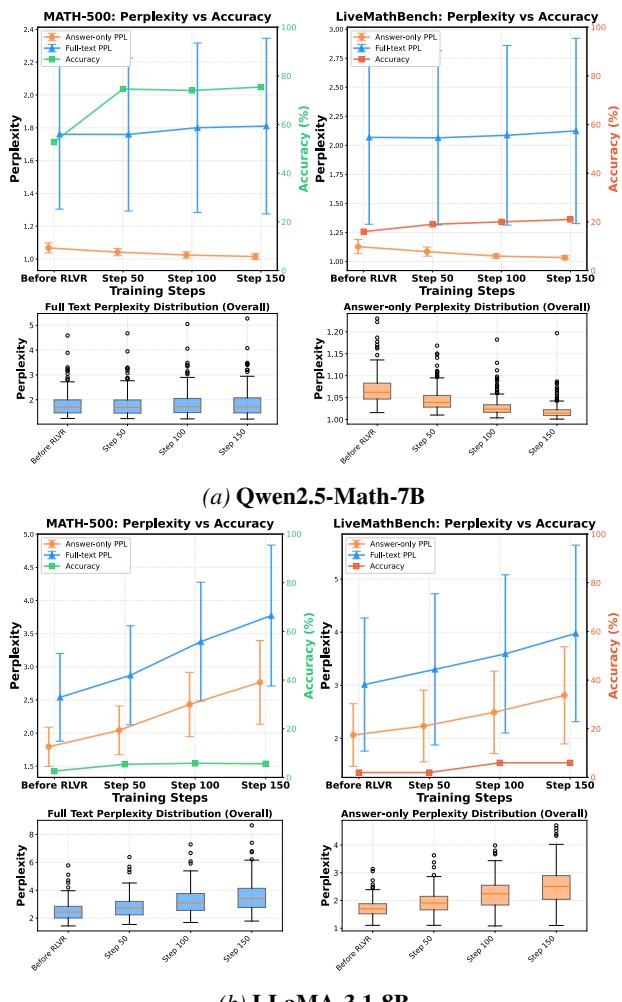

*(a)* **Qwen2.5-Math-7B**

*(b)* **LLaMA-3.1-8B**

*Figure 3.* **Perplexity Analysis.** (a) Qwen exhibits the Perplexity Paradox: rising full-text perplexity with falling answer perplexity on contaminated data (MATH-500), while both remain high on clean data (LiveMathBench). (b) LLaMA shows no such divergence, with uniformly rising perplexity across all conditions.

track this metric across incorrect rewards checkpoints at step 0 (base model), 50, 100, and 150.

Figure 3 reveals a **Perplexity Paradox** unique to Qwen2.5-Math: on contaminated datasets (MATH-500), answer-only perplexity *decreases* progressively while full-text perplexity *increases*. In contrast, both LLaMA and OLMo (Appendix D) exhibit *rising* perplexity in both metrics, indicating that spurious RLVR degrades their language modeling without activating memorization shortcuts. On leakage-free LiveMathBench, PPL remains at the highest level. This divergence constitutes a Perplexity Paradox: minimizing loss on specific memorized answers under spurious RLVR leads to degraded language modeling on input prompts, aligning with the "over-memorization" hypothesis (Ruan et al., 2025).

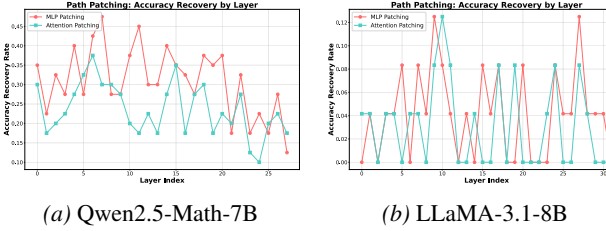

*(a)* Qwen2.5-Math-7B      *(b)* LLaMA-3.1-8B

*Figure 4.* **Path Patching Accuracy Recovery Comparison.** Left: Qwen exhibits a sustained high plateau followed by a sudden drop at L21, which marks the final injection of the correct answer and reveals a critical feature space divergence. Right: LLaMA shows no comparable recovery pattern and maintained extremely low recovery rates, confirming the absence of this memorization mechanism in the control model.

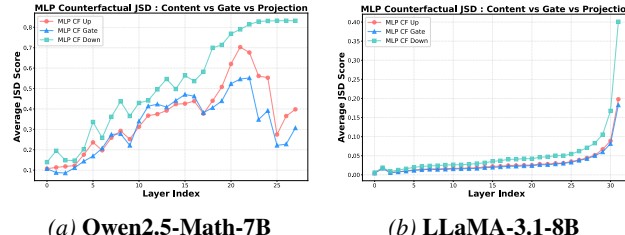

*(a)* **Qwen2.5-Math-7B**      *(b)* **LLaMA-3.1-8B**

*Figure 5.* **Layer-wise MLP Sub-component JSD Scores.** (a) In Qwen, JSD for $\mathbf{W}_{up}$ and $\mathbf{W}_{gate}$ peaks around Layer 21 and subsequently declines, identifying this depth as the locus of maximal parametric change. (b) In LLaMA, all components exhibit a monotonic increase, lacking the structural adaptation signature.

### 4.3. Locating the Functional Anchor: Static Analysis

Having established existence of memorization shortcuts, we now pinpoint their physical location within model's depth. We employ three complementary mechanistic probes:

**Path Patching: The Functional Cause.** To identify which components causally drive the correct output, we perform activation patching. We replace the activations of the base model with those from the spurious RLVR-tuned model at specific layers and measure the recovery of accuracy. Figure 4 reveals two critical findings specific to Qwen: (1) Patching the MLPs yields significantly higher accuracy recovery than patching attention heads, confirming that MLPs are the primary storage for this learned knowledge. (2) In Qwen, the accuracy recovery maintains a consistently high level across the erly layers, but it is the sustained high recovery at **Layers 18–20** immediately before the precipitous drop at Layer 21, combined with JSD scores beginning to climb at these layers shown in Figure 5, that highlights L18–L20 as the critical window where the effective signal is fully present immediately before the performance decline. In comparison, the LLaMA model shows no such significant recovery or layer-specific trends. This confirms that the specific memorization circuit identified in Qwen is not a general artifact of RLVR, but a symptom of activating contaminated knowledge. (See Appendix E for results on OLMo and Qwen3).

**JSD: The Structural Effect.** While Path Patching measures causal effect on accuracy, we employ a *counterfactual JSD* analysis to isolate each MLP sub-component's marginal contribution to the distributional shift. For the gated MLP architecture: $\text{MLP}(\mathbf{x}) = \big(\sigma(\mathbf{x}\mathbf{W}_{\text{gate}}) \odot \mathbf{x}\mathbf{W}_{\text{up}}\big)\mathbf{W}_{\text{down}}$, we perform the following counterfactual interventions: (1) $\mathbf{W}_{up}$ *contribution:* Replace $\mathbf{W}_{\text{up}}$ with its mean-value matrix $\bar{\mathbf{W}}_{\text{up}} = \mathbb{E}[\mathbf{W}_{\text{up}}] \cdot \mathbf{1}$ while keeping $\mathbf{W}_{\text{gate}}$ intact; (2) $\mathbf{W}_{gate}$ *contribution:* Replace $\mathbf{W}_{\text{gate}}$ with its mean-value matrix $\bar{\mathbf{W}}_{\text{gate}} = \mathbb{E}[\mathbf{W}_{\text{gate}}] \cdot \mathbf{1}$ while keeping $\mathbf{W}_{\text{up}}$ intact; (3) $\mathbf{W}_{down}$ *contribution:* Replace $\mathbf{W}_{\text{down}}$ with a norm-preserving ran-

dom matrix: $\mathbf{W}_{\text{rand}} = \frac{\|\mathbf{W}_{\text{down}}\|_F}{\|\mathbf{W}_{\text{init}}\|_F} \mathbf{W}_{\text{init}}, \quad \mathbf{W}_{\text{init}} \sim \mathcal{N}(0, \mathbf{I}).$

Figure 5 presents layer-wise counterfactual JSD scores. We observe distinct trajectories for sub-components in Qwen2.5-Math: divergence scores for both $\mathbf{W}_{\text{up}}$ and $\mathbf{W}_{\text{gate}}$ rise steadily to reach a **peak at Layers 21–22**, before declining in subsequent layers. In contrast, $\mathbf{W}_{\text{down}}$ divergence (representing total MLP output projection) maintains a sustained upward trend, converging to a high plateau. This pattern aligns with the established mechanistic role of MLPs, where $\mathbf{W}_{\text{up}}$ serves as a repository of candidate features (information vault) and $\mathbf{W}_{\text{gate}}$ acts as the selector. The JSD peak for these components at L21–L22 suggests the completion of specific knowledge injection: the model concentrates its parametric updates here to encode the memorized features.

Crucially, this structural insight explains the divergence observed in our Path Patching results. While the internal components ($\mathbf{W}_{\text{up/gate}}$) gradually decreased after L22, the output projection ($\mathbf{W}_{\text{down}}$) exhibits *persistent high values* across all subsequent layers. This indicates that L21–L22 act as **Structural Adapters** that permanently rotate the feature space. Consequently, although the MLP continues to function with high confidence (high JSD), its output resides in a new manifold that is structurally incompatible with the base model's remaining layers, leading to the observed degradation in patching accuracy (L21+). In contrast, LLaMA-3.1-8B exhibits monotonically increasing JSD across all components without the characteristic peak-reversal pattern. This monotonic trend confirms that spurious RLVR fails to induce localized structural adaptation in models lacking pre-existing contaminated knowledge. (See Appendix F for OLMo and Qwen3 results, which show similar behavior).

**Logit Lens: The Semantic Transition.** Having localized the specific layers responsible for knowledge injection and structural adaptation, we now examine how these internal mechanisms materialize in the token space. To investigate this, we conduct a controlled case study using a obvious leakage sample from MATH-500. We sample two generation trajectories for the same prompt with a temperature of

0.7: one where the model successfully retrieves the correct answer ("4"), and one where it fails ("3") (See Appendix J).

In the successful trajectory (Figure 6 Top), a clear activation sequence appears. The MLP at Layer 19 first introduces a precursor signal. Following a temporary probability drop at Layers 21–22: which aligns with the feature space transformation observed in our JSD analysis, i.e., the MLP at Layer 23 aggressively injects the correct answer token "4". The residual stream successfully integrates this signal, stabilizing on "4" until the final output. The failure case (Figure 6 Bottom) provides a critical counterfactual. Even though the model eventually outputs the wrong answer "3", the heatmap of Figure 20 in Appendix G reveals that the MLPs at Layers 23–25 *still* attempt to inject the correct answer "4". However, the initial priming signal at Layer 19 is notably weaker compared to the successful run. Without sufficient intensity from these middle layers to override the stream's inertia, the subsequent MLP injections fail to correct the trajectory. This confirms that while the MLPs persistently store the memorized solution, acting as a fixed memory store, in light of the conclusions of the aforementioned experiments, the successful retrieval of this knowledge is causally determined by the signal strength established in these middle layers which might be in the L18–L20 region.

## 4.4. Tracking the Bifurcation: Dynamic Analysis

To mathematically validate the *Functional Anchor* hypothesis, we extend our analysis from discrete layer-wise probes to continuous dynamic modeling using **Neural Differential Equations (NDEs)**. We model main residual stream $H = \{h_0, \ldots, h_L\}$ as a continuous trajectory $h(t)$ governed by $\frac{dh(t)}{dt} = f_\phi(h(t), t)$, where $f_\phi$ is parameterized by a two-layer MLP trained to approximate discrete updates.

**Trajectory Separation.** Figure 7a visualizes the PCA-projected trajectories of leakage samples versus stable samples. The two trajectories overlap substantially in early layers but exhibit a distinct **bifurcation** in the middle layers, subsequently diverging into separate regions of the latent space. This visual evidence confirms that the processing of memorized and reasoned information follows fundamentally different dynamic paths after the anchor layers.

**Separation Force and Velocity.** We quantify this bifurcation using two metrics derived from the NDE framework: (1) **Separation Force:** Defined as $\left\| \frac{dh_{\text{leak}}}{dt} - \frac{dh_{\text{stable}}}{dt} \right\|$, measuring the instantaneous divergence in update directions between leakage and stable samples; (2) **Velocity Difference:** Defined as $\|\|v_{\text{leak}}\| - \|v_{\text{stable}}\|\|$ where $v = \frac{dh}{dt}$, measuring the difference in update magnitudes. Figure 7b (Left) shows that the Separation Force peaks precisely at Layers 18, 19, and 20, mathematically confirming that these layers exert the strongest directional influence to steer leakage samples onto the memorization trajectory. Conversely, Figure 7b (Right)

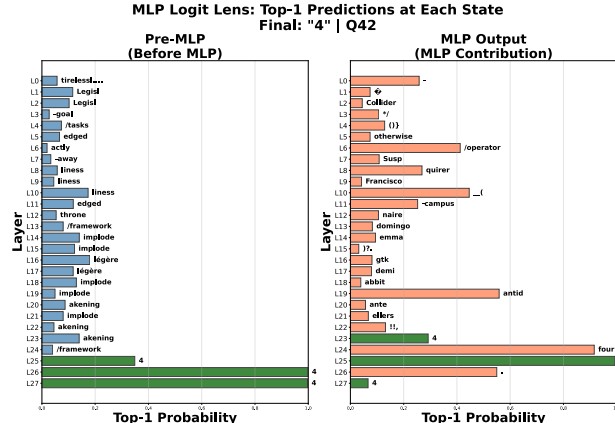

*(a)* Successful Retrieval Trajectory (Output: Correct Answer "4")

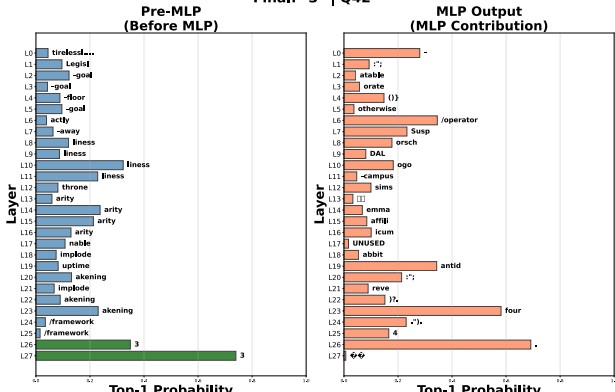

*(b)* Failed Retrieval Trajectory (Output: Wrong Answer "3")

*Figure 6.* **Logit Lens Analysis. (Top)** In successful case, the Functional Anchor (L19) primes the stream, and after structural adaptation (L21-22), the MLP at L23 successfully injects the correct answer "4". **(Bottom)** In the failure case, despite the MLP still attempting to inject "4" at L23-25 (visible in heatmap of Figure 20), the weaker initial signal from the Anchor allows the residual stream to drift towards the incorrect token "3". This confirms that MLPs store the memory, and will continue to output memories even in the presence of interference.

shows that the velocity difference is minimal at L19 but increases in later layers. This pattern is consistent with our Structural Adapter interpretation: the later layers amplify the signal magnitude (velocity) without substantially altering its direction (force), thereby propagating the decision made at the *Functional Anchor* layers.

**Linear Probing Validation.** As an independent validation, we train linear probes on each layer's residual stream to classify leakage versus stable samples. Figure 7c shows AUC-ROC score peaking at Layer 20. This confirms that information distinguishing memorization from reasoning becomes maximally linearly separable immediately after *Functional Anchor* layers have performed their critical signal injection.

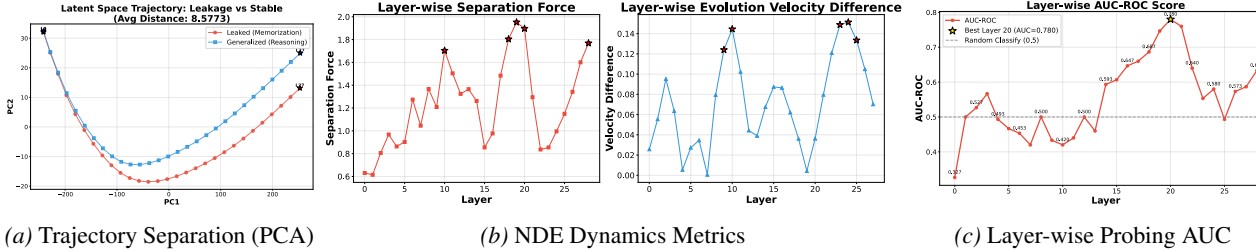

*(a)* Trajectory Separation (PCA)      *(b)* NDE Dynamics Metrics      *(c)* Layer-wise Probing AUC

*Figure 7.* **Bifurcation Analysis via Multiple Lenses. (Left)** PCA-projected latent space trajectories show clear bifurcation between leakage (red) and stable (blue) samples in middle layers. **(Middle)** Separation Force peaks at L18–L20 (identifying the causal origin), while Velocity Difference increases in later layers (reflecting signal amplification). **(Right)** Linear probe AUC-ROC peaks at Layer 20, confirming maximal separability between memorization and reasoning modes immediately after the *Functional Anchor*.

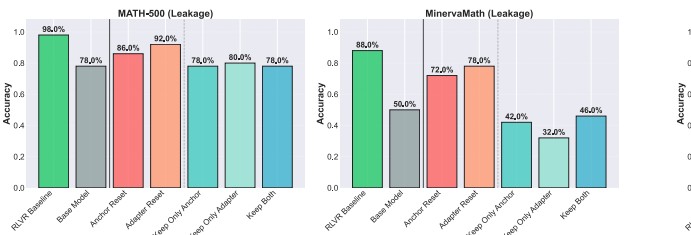
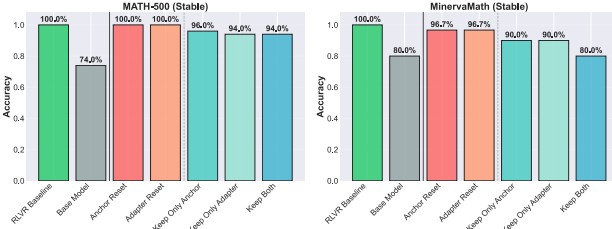

*(a)* Overall accuracy on leakage samples across ablation conditions. Anchor reset and Adapter reset cause moderate drops.

*(b)* Overall accuracy on stable samples. Performance remains stable under reset conditions.

*Figure 8.* **Ablation study results.** Leakage samples from contaminated datasets exhibit higher sensitivity to layer manipulation than stable samples. The leakage-free LiveMathBench shows a different pattern, which are provided in Appendix H.

## 5. Ablation Study

To validate our findings regarding the *Functional Anchor* and *Structural Adapter*, we conduct systematic ablation experiments on the RLVR-tuned model. These experiments are designed to verify both the *necessity* and *sufficiency* of these layer groups for memorization. More results for other datasets like LiveMathBench are provided in Appendix H.

We design two complementary groups of experiments on the RLVR-tuned model: **Group 1: Necessity Tests (Reset).** We selectively replace specific layer groups with base model weights: (1) *Anchor Reset*: Replace L18–20 MLP weights with base model weights; (2) *Adapter Reset*: Replace L21–22 MLP weights with base model weights. **Group 2: Sufficiency Tests (Keep Only).** We retain only specific RLVR-tuned layers while resetting all others to base: (1) *Keep Only Anchor*: Only L18–20 retain RLVR weights; (2) *Keep Only Adapter*: Only L21–22 retain RLVR weights; (3) *Keep Both*: L18–22 retain RLVR weights.

Figure 8 shows several key findings: **Anchor layers are more critical than Adapter layers.** Resetting the Anchor causes leakage accuracy to drop by more than 10% on contaminated datasets (MATH-500: 98%→86%; MinervaMath: 88%→72%), while Adapter Reset produces smaller drops ($\Delta = -6\%$ and $-10\%$). This confirms the Anchor layers serve as primary storage for memorized knowledge.

The remaining 86% accuracy after Anchor Reset is also informative. It does not imply that L18–20 alone store all contaminated answers; instead, it shows that the shortcut is anchor-triggered but partially distributed. On the MATH-500 leakage subset, the base model already reaches 78%, so replacing L18–20 with base weights preserves some pre-existing contamination while removing most RLVR-only gain (98%→86%). In contrast, keeping only the Adapter reaches 80%, and randomizing the Anchor collapses accuracy to 34%, far below both base and RLVR. Thus the Anchor is best interpreted as the integrity-critical retrieval trigger, while downstream layers retain residual shortcut-compatible signal.

**Neither layer group alone is sufficient for memorization.** Keep Only experiments show that isolated layer groups fail to recover RLVR performance. On MinervaMath, Keep Only Adapter yields only 32% accuracy (vs. 50% base model), while Keep Only Anchor reaches 42%, and Keep Both improves to 46%. This demonstrates that Anchor and Adapter layers form a collaborative circuit for memory retrieval.

**Contaminated and clean datasets exhibit divergent sensitivity.** On contaminated datasets, leakage samples show high sensitivity to Anchor manipulation. Conversely, on leakage-free LiveMathBench, accuracy remains stable at 70% across all ablation conditions (Appendix H), provid-

ing direct evidence that Anchor layers encode *dataset-contaminated knowledge* rather than general reasoning.

**Stable samples are robust to layer manipulation.** Across all reset experiments, stable sample accuracy remains nearly identical to the RLVR baseline. This indicates genuine reasoning capabilities are distributed across the broader network and do not critically depend on these specific layers, reinforcing that Anchor layers are specialized for memorization shortcuts.

We further check whether this localization is tied only to the 7B checkpoint by running a supplementary Qwen2.5-Math-1.5B analysis with the same Anchor (L18–20) and Adapter (L21–22) partition. Although the smaller model has a weaker leakage signal overall, the same qualitative pattern appears on its 38-case MATH-500 leakage subset: Anchor Reset drops accuracy from 39.5% to 2.6%, Keep Only Adapter falls to 0%, and the largest mean FFN drop occurs at L18. Path patching is again MLP-dominant, with FFN AUC 0.0789 versus attention AUC 0.0498 (paired sign-flip $p = 6.99965 \times 10^{-4}$). We therefore frame the claim as within-family evidence rather than universality: when a Qwen2.5 model exhibits a clear leakage signal, the same Anchor-centered mechanism remains visible across scales.

## 6. Mechanistic Intervention

Building on our previous findings, we investigate whether targeted intervention on these layers can suppress this undesired behavior. We design a neuron-level steering method that directly manipulates MLP activations during inference to disrupt memorization circuit without model retraining.

### 6.1. Task-Relevant Neuron Identification

In Qwen2.5-Math MLP layers, each neuron functions as a gated memory unit. For a hidden state $\mathbf{x}_\ell \in \mathbb{R}^{d_{\text{hidden}}}$ at layer $\ell$, the MLP computation proceeds as: $\mathbf{k}_\ell = \text{SiLU}(\mathbf{x}_\ell \mathbf{W}_{\text{gate}}) \odot (\mathbf{x}_\ell \mathbf{W}_{\text{up}}) \in \mathbb{R}^{d_{\text{mlp}}}$, and $\text{MLP}(\mathbf{x}_\ell) = \mathbf{k}_\ell \mathbf{W}_{\text{down}} = \sum_{i=1}^{d_{\text{mlp}}} k_\ell^{(i)} \cdot \mathbf{w}_i^{\text{down}}$, where $d_{\text{mlp}} \gg d_{\text{hidden}}$ (e.g., $14336 \gg 3584$ in Qwen2.5-Math-7B), and $\mathbf{w}_i^{\text{down}} \in \mathbb{R}^{d_{\text{hidden}}}$ is the $i$-th *row* of $\mathbf{W}_{\text{down}} \in \mathbb{R}^{d_{\text{mlp}} \times d_{\text{hidden}}}$. Each scalar $k_\ell^{(i)}$ acts as a key that gates contribution of its corresponding neuron value $\mathbf{w}_i^{\text{down}}$ to output. We project each neuron's value vector into vocabulary space via the unembedding matrix: $\mathbf{v}_i = \text{LayerNorm}(\mathbf{w}_i^{\text{down}})\mathbf{W}_U \in \mathbb{R}^{|V|}$. We then compute a relevance score combining key activation magnitude and semantic overlap with answer tokens: $s_i = |k_\ell^{(i)}| \cdot \left(1 + \lambda \cdot \frac{|\text{Top-10}(\mathbf{v}_i) \cap \text{Tokens}(y)|}{|\text{Tokens}(y)|}\right)$, where Top-10($\mathbf{v}_i$) denotes the 10 highest-probability tokens after softmax, $y$ is the ground-truth answer, and $\lambda$ is used to control the intensity of semantic alignment. We select the top-$k$ neurons ($k = 10$) ranked by these scores as the intervention

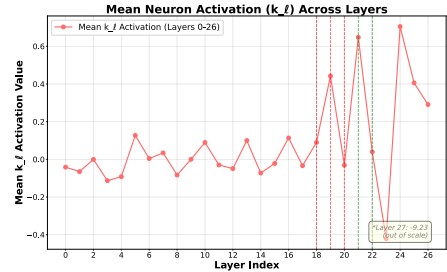

*Figure 9.* **Mean Neuron Activation Across Layers.** Identified task-relevant neurons exhibit maximal activation at L19, 21, 24.

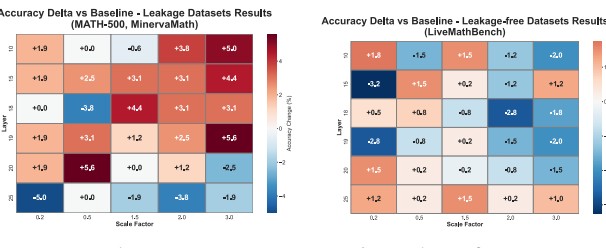

*(a) Leakage Dataset*      *(b) Leakage-free Dataset*

*Figure 10.* **Dataset-Level Steering: Accuracy Change. (a)** On leakage datasets, Layer 18 exhibits maximal sensitivity ($\pm 4\%$). **(b)** On leakage-free dataset, steering produces no systematic pattern, confirming that intervention specifically targets contamination-dependent circuits rather than general reasoning pathways.

target set $\mathcal{N}_\ell^{\text{task}}$.

### 6.2. Causal Steering

We apply multiplicative scaling to selected neurons' key activations: $\hat{k}_\ell^{(i)} = \{\alpha \cdot k_\ell^{(i)} \text{ for } i \in \mathcal{N}_\ell^{\text{task}}; \text{ else } k_\ell^{(i)}\}$, where $\alpha$ is scaling factor, applied uniformly across all generation steps. To validate our neuron selection strategy, we first examine activation magnitudes $k_\ell$ of the identified task-relevant neurons across layers. Figure 9 reveals that these neurons exhibit peak activations precisely at L19, L21, and L24 that align with both our *Functional Anchor* hypothesis and the MLP injection patterns observed in Logit Lens analysis (Figure 6). This convergence of evidence confirms that our selection method successfully identifies neurons that are mechanistically central to memorization circuit.

### 6.3. Results

Figure 10 presents accuracy change when applying neuron-level steering across representative layers on full leakage datasets. Layer 18 exhibits maximal sensitivity ($-3.8\%$ under suppression, $+4.4\%$ under amplification), consistent with its role as *Functional Anchor*. In contrast, Layer 25 shows uniform degradation across all factors, reflecting its role as a Structural Adapter where arbitrary manipulation disrupts transformation process. Critically, on leakage-free dataset, steering yields no systematic pattern, con-

*Table 1.* **Probe-gated shortcut audit.** The conditional intervention fires on contamination-sensitive MATH-500 cases but remains inactive on fresh benchmarks. Accuracies are reported before and after the gated intervention.

| Split | Gate | RLVR | Conditional |
|---|---|---|---|
| MATH-500 `wrong→right` | 79/128 | 100.0% | 85.16% |
| AIME-2025 | 0/30 | 30.0% | 30.0% |
| LiveMathBench | 0/100 | 21.0% | 21.0% |
| OlymMATH-en-easy(OOD) | 0/100 | 7.0% | 7.0% |
| OmniMATH100-rule(OOD) | 3/100 | 36.0% | 35.0% |

firming that intervention specifically targets contamination-dependent circuits rather than general reasoning pathways.

To elucidate the fine-grained pathway through which neuron scaling modulates memorization retrieval, we conduct layerwise trajectory analysis on individual leakage samples in Figure 24a of Appendix I. Under amplification ($\alpha = 3.0$), a special formatting token reaches top-1 at Layer 22 one layer earlier, and the target answer token achieves higher probability. Conversely, suppression ($\alpha = 0.2$) delays trigger activation and reduces answer token probability. This bidirectional control over both the *timing* and *strength* of retrieval provides direct mechanistic evidence that the *Functional Anchor* neurons constitute a causally sufficient pathway for memory activation. Beyond gradual modulation, we observe a more striking phenomenon in cases where baseline retrieval fails entirely as shown in Figure 24b of Appendix I. For Question 213 (ground truth: "2"), both suppression and baseline produce incorrect outputs with uniformly low token probabilities. However, amplification triggers a qualitatively different trajectory: starting from the middle layers, a dominant token maintains high probability, while the correct answer "2" emerges abruptly only in the final two layers. This suggests that amplification *activates a latent shortcut pathway* inaccessible to the base model.

### 6.4. Probe-Gated Shortcut Audit

The global steering results above establish causal controllability, but a practical audit should not suppress the circuit on every prompt. We therefore add a selective variant that first scores each prompt with the Layer-19 leakage probe and intervenes only when the score exceeds a conservative threshold calibrated at the 99th percentile of negative-control prompts. When triggered, we apply the same inference-time mechanism at the identified circuit, using Layer 19, the top-50 probe-ranked neurons, and a suppression factor $\alpha = 0.5$. This turns the intervention into a counterfactual test: if an RLVR success disappears only under the gated intervention, the success is shortcut-dependent; if the gate stays inactive, the benchmark result is left unchanged.

Table 1 shows that the gate activates on 61.7% of the MATH-500 `wrong→right` subset and removes 19 RLVR-only

successes, reducing accuracy from 100.0% to 85.16% (exact McNemar $p = 3.81 \times 10^{-6}$). In contrast, the gate fires on 0/30 AIME-2025 and 0/100 LiveMathBench examples, so conditional accuracy remains identical to the RLVR checkpoint. The same pattern holds on additional recent benchmarks: OlymMATH-en-easy is unchanged, and OmniMATH100-rule changes by only one point with a nonsignificant paired test ($p = 1.0$). Thus, the intervention is best understood as an offline shortcut-dependence audit rather than a blanket reasoning improver: it selectively stress-tests suspicious gains while preserving fresh-benchmark behavior.

## 7. Conclusion

In this work, we present a mechanistic analysis of how spurious RLVR induces stubborn memorization in the Qwen2.5-Math models. We identify the Perplexity Paradox: rising prompt perplexity alongside falling answer perplexity, as a macroscopic fingerprint of this phenomenon, where general language coherence is sacrificed for shortcut learning. Mechanistically, we uncover a fundamental decoupling between function and structure: utilizing Path Patching, Logit Lens, JSD analysis, and Neural Differential Equations (NDEs), we pinpoint Layers 18–20 as the **Functional Anchor** that causally injects the memorization trigger, while Layers 21+ serve as **Structural Adapters** that undergo drastic weight reorganization to accommodate this signal. Ablation experiments confirm that these layers encode dataset-specific shortcuts rather than general reasoning improvements, as manipulation selectively degrades contaminated benchmarks while leaving clean benchmarks unaffected. Finally, we demonstrate that scaling specific MLP keys within this circuit allows for bidirectional causal steering, i.e., artificially amplifying or suppressing contamination-driven performance. With a conservative probe gate, the same intervention becomes a selective shortcut audit that suppresses suspicious contaminated gains while preserving behavior on fresh benchmarks. Our findings provide mechanistic grounding for understanding RLVR vulnerabilities and open avenues for targeted mitigation strategies.

### Acknowledgments

This work has received funding from the European Union's Horizon Europe research and innovation programme project TrustLLM under grant agreement No 101135671.

### Impact Statement

This work presents a mechanistic investigation into how reinforcement learning can inadvertently trigger shortcuts by activating latent memorized data rather than fostering genuine reasoning. The implications of our findings are

twofold:

1. **Strengthening AI Trustworthiness and Evaluation**: Our discovery of the Anchor-Adapter circuit and the Perplexity Paradox provides new tools for the AI community to distinguish between appearance of capability and genuine generalization. As LLMs are increasingly deployed in critical domains like mathematics, science, and coding, it is essential that their performance is rooted in robust reasoning. By providing a methodology to localize and steer these shortcuts, our work helps developers identify and mitigate the effects of data contamination, leading to more honest and reliable AI systems.

2. **Mitigating the Risks of Spurious Optimization**: We highlight a critical vulnerability in RLVR: the potential for models to optimize for reward signals through stubborn memorization. This underscores the need for more sophisticated reward functions and evaluation benchmarks that are resistant to leakage. Furthermore, our causal steering techniques offer a path toward decontaminating models post-training, reducing the risk of deploying models that appear performant but fail on novel, out-of-distribution tasks.

**Ethical Considerations**: While our work provides tools to detect and suppress memorization, the same techniques could theoretically be used to amplify a model's reliance on specific datasets. We advocate for the responsible use of mechanistic interpretability tools to ensure they are used to enhance transparency and safety rather than to mask architectural flaws.

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

## A. Code

We release our code: https://github.com/idwts/How-RLVR-Activates-Memorization-Shortcuts.

## B. Preliminaries

Our work builds upon the emerging understanding of RLVR and the specific vulnerabilities identified in Qwen2.5 architecture. Recent findings indicate that while RLVR aims to reinforce robust reasoning paths (Shao et al., 2025), in models like Qwen2.5, it can inadvertently exploit data contamination to trigger memorization rather than generalization (Wu et al., 2025b). To dissect the internal mechanisms driving this phenomenon, we ground our methodology in mechanistic interpretability (Conmy et al., 2023). We focus on Transformer decoder-only architectures, modeling the propagation of information through the residual stream to identify the physical location of memory activation.

**Multilayer Perceptron (MLP).** In Transformer architectures, MLPs are widely regarded as the primary storage units for parametric knowledge (Geva et al., 2021; Kim et al., 2024). Each MLP layer functions as a key-value memory, where the input vector acts as a query to activate specific neurons ("keys"), which then retrieve corresponding "values" to update the residual stream. In the context of our study, MLPs are the critical components of interest, as they are hypothesized to store and retrieve the specific memorized solutions triggered by spurious RLVR.

**Path Patching.** To causally attribute the model's output to specific internal components, we employ Path Patching (Meng et al., 2022). This technique allows us to measure the importance of specific edges in the computational graph (e.g., from a specific layer's output to the final logits) by intervening on activations. In our context, we swap activations between *Leakage samples* (where memorization is activated) and *Stable samples* (where genuine reasoning is maintained). By observing which layer interventions successfully restore the memorized output in the stable context, we can isolate the specific components causally responsible for triggering the memory retrieval shortcut.

**Logit Lens.** To visualize the intermediate decoding process, we utilize the Logit Lens technique (nostalgebraist, 2020). This method projects hidden states $\mathbf{x}_\ell$ from intermediate layer $\ell$ directly into the vocabulary space using the model's pre-trained unembedding matrix $\mathbf{W}_U$. Formally: $\text{LogitLens}(\mathbf{x}_\ell) = \text{LayerNorm}(\mathbf{x}_\ell)\,\mathbf{W}_U$. This allows us to track the layer-wise emergence of the target answer tokens, revealing exactly when the model commits to a memorized answer before the final layer.

**Jensen-Shannon Divergence (JSD).** To rigorously quantify the alignment between intermediate hidden states and the model's final decision, we employ the Jensen-Shannon Divergence (JSD). Building on the Logit Lens projection, we define $q(x_\ell) = \text{softmax}\big(\text{LogitLens}(x_\ell)\big)$ as the vocabulary distribution derived from layer $\ell$, and $P_{\text{final}}$ as the model's final output distribution. The divergence score is calculated as $\text{JSD}\big(q(x_\ell) \parallel P_{\text{final}}\big) = \frac{1}{2}\text{KL}\big(q(x_\ell) \parallel M\big) + \frac{1}{2}\text{KL}\big(P_{\text{final}} \parallel M\big)$, where $M = \frac{1}{2}\big(q(x_\ell) + P_{\text{final}}\big)$ represents the average distribution and KL denotes the Kullback-Leibler divergence. Similar to the leave-one-out strategy in the context attribution setting (Cohen-Wang et al., 2024; Li et al., 2025b), a high JSD score indicates that the intermediate representation at layer $\ell$ differs substantially from the final output distribution, suggesting that significant computational transformations occur in subsequent layers.

**Neural Differential Equations (NDEs).** To formalize the continuous evolution of hidden states, we adopt the framework of Neural Ordinary Differential Equations (Neural ODEs) (Chen et al., 2018). Since residual connections in Transformers resemble the Euler discretization of a continuous dynamical system, we model the layer-wise trajectory of hidden states as a continuous flow. Specifically, the residual block $\mathbf{x}_{\ell+1} = \mathbf{x}_\ell + \mathcal{F}_\theta(\mathbf{x}_\ell)$ can be viewed as one step of Euler integration with step size $h = 1$ for the ODE: $\frac{\mathrm{d}\mathbf{x}(t)}{\mathrm{d}t} = \mathcal{F}_\theta\big(\mathbf{x}(t)\big)$. This continuous formulation allows us to compute the *separation force* acting on latent representations and to mathematically pinpoint the bifurcation point where the trajectory of a memorized sample physically diverges from that of a reasoning sample. In our implementation, we approximate $\mathcal{F}_\theta$ using a two-layer MLP fitted to the discrete layer-wise updates $\Delta\mathbf{x}_\ell = \mathbf{x}_{\ell+1} - \mathbf{x}_\ell$.

## C. Partial Prompt Evaluation Details

This appendix provides the complete results for the Partial Prompt Evaluation discussed in Section 3.1. This experiment assesses whether models possess pre-existing knowledge of the ground-truth answers by prompting them with the question text alone (without Chain-of-Thought) to see if they can complete the exact answer string.

**Cross-Model Accuracy Comparison (Figure 11).** To provide context for the Partial Prompt Evaluation, we first present

**Model Accuracy Comparison: Before vs After RLVR Fine-tuning**

*Figure 11.* **Accuracy Comparison Across Models and Datasets.** Qwen3-8B shows high baseline accuracy and dramatic improvement under spurious RLVR, while LLaMA-3.1-8B and OLMo-2-1124-7B remain at a low level performance regardless of training.

the overall accuracy of Qwen3-8B, LLaMA-3.1-8B, and OLMo-2-1124-7B across six benchmarks before and after spurious RLVR training with incorrect rewards. A striking dichotomy emerges: while LLaMA and OLMo maintain consistently low accuracy regardless of RLVR training, Qwen3-8B exhibits both high baseline performance and substantial gains after spurious RLVR. This pattern mirrors the behavior observed in Qwen2.5-Math-7B and provides strong *prima facie* evidence that Qwen3-8B also suffers from data contamination, which spurious RLVR subsequently activates.

**Qwen2.5-Math-7B (Figure 12).** On contaminated datasets (MATH-500, MinervaMath), the "Wrong→Right" group (green line) shows a dramatic increase in both ROUGE-L and exact completion accuracy after RLVR. This confirms that the model is learning to retrieve specific memorized strings.

**Qwen3-8B (Figure 13).** Consistent with Qwen2.5-Math-7B, Qwen3-8B exhibits increasing ROUGE-L scores and completion accuracy as the prompt ratio increases, confirming its ability to retrieve memorized answers. However, the separation between the pre- and post-RLVR curves is less pronounced compared to Qwen2.5-Math-7B, particularly in the "Wrong→Right" group. This suggests that the extent of data leakage or the degree to which spurious RLVR activates these memorized answers is less significant in Qwen3-8B, indicating a potentially lower reliance on contamination or weaker memory activation under spurious RLVR conditions.

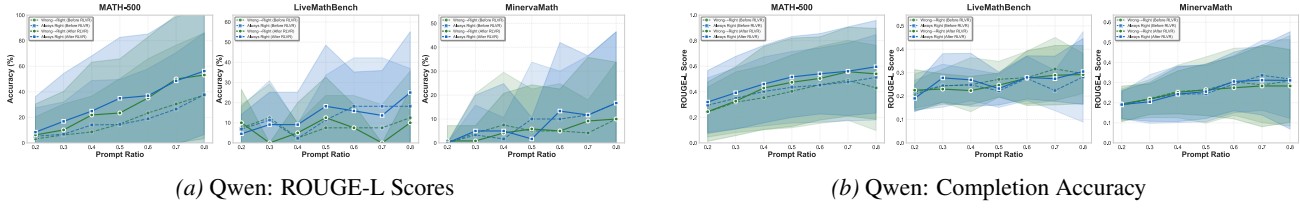

*(a)* Qwen: ROUGE-L Scores             *(b)* Qwen: Completion Accuracy

*Figure 12.* **Qwen2.5-Math-7B Partial Prompt Results.** The sharp rise in the "Wrong→Right" group (green) on contaminated datasets signals the activation of memorized answers.

**Control Models (Figure 14).** In contrast, for LLaMA-3.1-8B and OLMo-2-1124-7B, both ROUGE-L and completion accuracy remain at a consistently low level. Crucially, these metrics show no significant correlation or discernible trend as the prompt ratios increases, confirming the absence of pre-existing contamination.

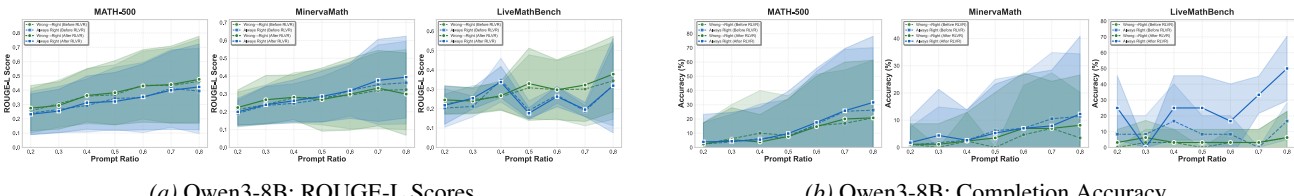

*(a)* Qwen3-8B: ROUGE-L Scores

*(b)* Qwen3-8B: Completion Accuracy

*Figure 13.* **Qwen3-8B Partial Prompt Results.** Qwen3-8B shows the expected increase in ROUGE-L and completion accuracy as prompt ratios increase, but the separation between pre- and post-RLVR is less pronounced compared to Qwen2.5-Math-7B, suggesting weaker memory activation under spurious RLVR.

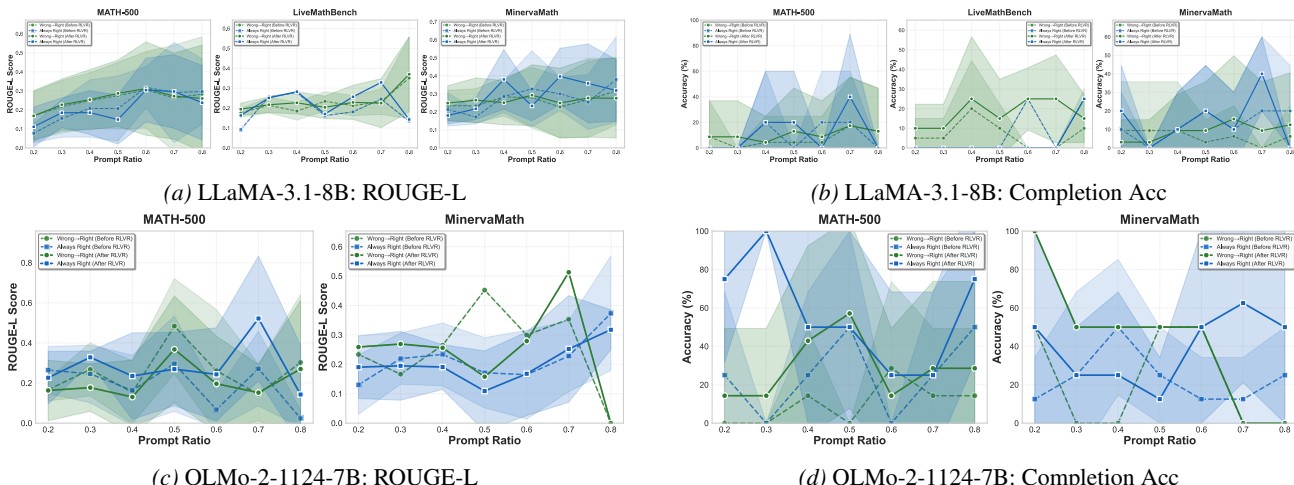

*(a)* LLaMA-3.1-8B: ROUGE-L

*(b)* LLaMA-3.1-8B: Completion Acc

*(c)* OLMo-2-1124-7B: ROUGE-L

*(d)* OLMo-2-1124-7B: Completion Acc

*Figure 14.* **Control Models Partial Prompt Results.** Neither LLaMA (top) nor OLMo (bottom) shows significant improvement in completion capability under spurious RLVR, confirming no memorization activation occurred. Note that for OLMo-2-1124-7B, there are no samples categorized as "Wrong→Right" or "Stable" on the LiveMathBench.

## D. Complete Perplexity Analysis

This appendix provides the complete perplexity heatmaps, along with results for the OLMo-2-1124-7B control model.

**Heatmap Visualization.** Figure 15 presents the heatmap showing perplexity across all six benchmarks and four training checkpoints. This visualization clearly shows the dichotomy between contaminated datasets (MATH-500, AIME, MinervaMath, etc.) where answer perplexity drops, and clean datasets (LiveMathBench) where it remains high.

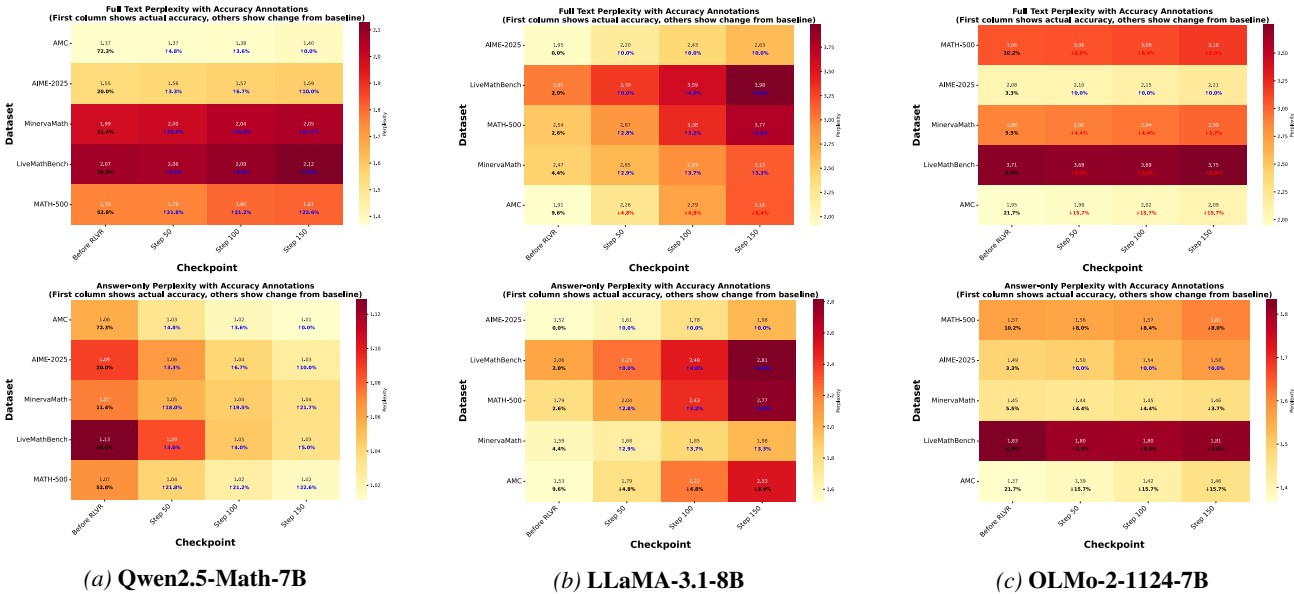

*(a)* **Qwen2.5-Math-7B**      *(b)* **LLaMA-3.1-8B**      *(c)* **OLMo-2-1124-7B**

*Figure 15.* **Compact Heatmap Comparison Across Three Models.** Full-text (top) and answer-only (bottom) perplexity for each model. Percentage annotations show base accuracy and improvement. Only Qwen exhibits the characteristic answer perplexity drop on contaminated datasets.

**OLMo-2-1124-7B Analysis.** Figure 16 presents the perplexity analysis for OLMo-2-1124-7B. Consistent with LLaMA, OLMo exhibits no Perplexity Paradox. Both full-text and answer-only perplexity increase uniformly across all datasets. This confirms that spurious RLVR fails to activate memorization shortcuts in models without pre-existing data contamination.

## E. Additional Path Patching Results

Complementing the main text comparison, Figure 17a presents the path patching results for the second control model, OLMo-2-1124-7B. Similar to LLaMA, OLMo exhibits no significant accuracy recovery or distinctive layer-wise trends when patching activations from the RLVR-tuned model back to the base model. Notably, Qwen3-8B also fails to display the characteristic middle-layer peak observed in Qwen2.5-Math-7B (Figure 4a). Instead, its recovery pattern similar to the control models LLaMA and OLMo, further corroborates our previous findings (Appendix C) that its data leakage and memory activation under spurious RLVR are significantly less pronounced compared to Qwen2.5-Math-7B.

## F. Additional JSD Results

We present the layer-wise MLP sub-component JSD scores for OLMo-2-1124-7B and Qwen3-8B in Figure 18. For OLMo (Left), all three JSD components exhibit a monotonic increase across layers as with LLaMA. There is no peak or decline as observed in Qwen, indicating that OLMo does not develop the kind of localized structural adaptation associated with memorization triggers. In contrast, Qwen3-8B (Right) replicates the distinctive structural signature observed in Qwen2.5-Math-7B: both $\mathbf{W}_{\text{up}}$ and $\mathbf{W}_{\text{gate}}$ exhibit a clear peak followed by a subsequent decline, with the peak occurring at **Layer 33** . Meanwhile, $\mathbf{W}_{\text{down}}$ maintains a sustained high divergence in Qwen3-8B, consistent with its role as the output projection layer. This architectural consistency across different Qwen family models strongly suggests that the circuit is not an artifact of a specific model checkpoint, but rather a systematic property of how the Qwen architecture responds to spurious RLVR when operating on contaminated data.

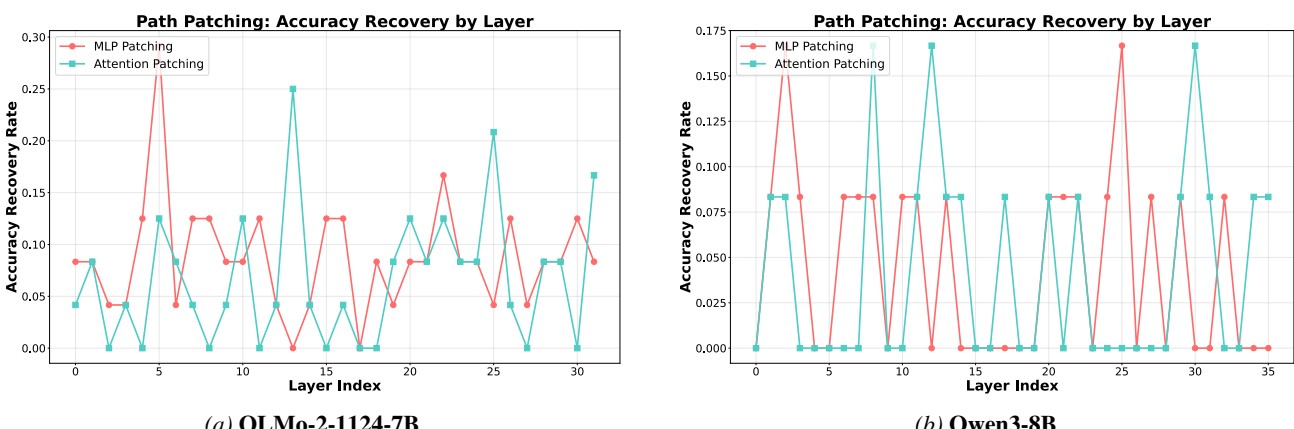

*Figure 16.* **Perplexity Analysis for OLMo-2-1124-7B.** Layout identical to Figure 3. OLMo shows no divergence pattern, with uniformly rising perplexity and minimal accuracy changes across all conditions.

*(a)* **OLMo-2-1124-7B**                    *(b)* **Qwen3-8B**

*Figure 17.* **Path Patching Recovery for Additional Models.** (a) Consistent with LLaMA, OLMo does not display the middle-layer signal injection pattern observed in Qwen2.5-Math-7B. (b) Similarly, Qwen3-8B has a weaker contamination effects compared to Qwen2.5-Math-7B.

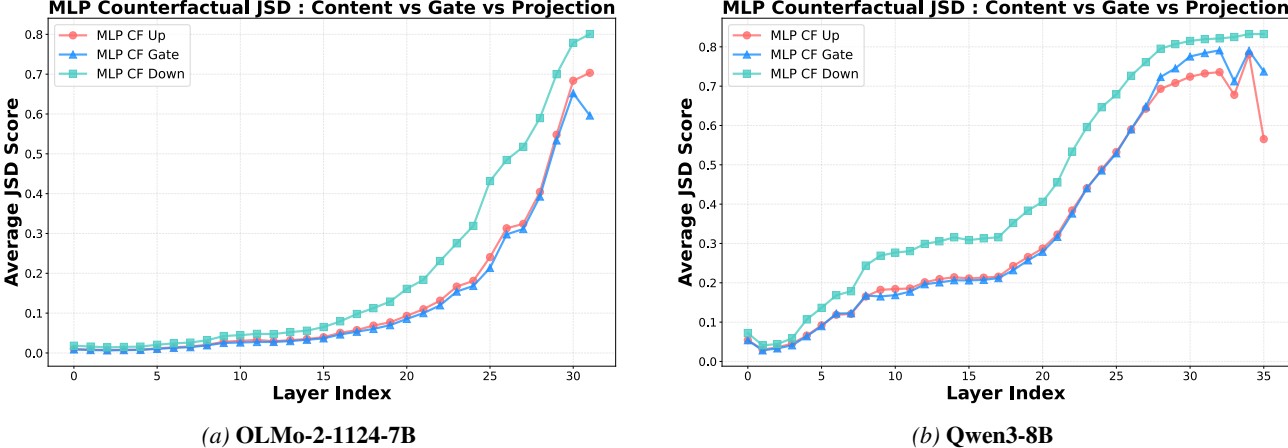

*(a)* **OLMo-2-1124-7B**        *(b)* **Qwen3-8B**

*Figure 18.* **Layer-wise MLP Sub-component JSD Scores for Additional Models.** (a) OLMo exhibits monotonic increase across all components, consistent with non-contaminated control models. (b) Qwen3-8B replicates the peak-and-decline pattern at Layer 33, confirming the generalizability of the Structural Adapter mechanism within the Qwen architecture family.

## G. Additional Results on Logit Lens

To further validate that the memorization activation is specific to the leakage activation observed in Qwen2.5-Math, we perform the same Logit Lens analysis on our control models, LLaMA-3.1-8B and OLMo-2-1124-7B, as well as on additional test cases for Qwen.

**Control Models (LLaMA & OLMo).** As shown in Figure 19, the internal dynamics of the control models differ fundamentally from Qwen. Neither LLaMA nor OLMo exhibits the characteristic peak-valley-rise pattern or the aggressive MLP signal injection in the middle layers. Instead, the probability of the answer token remains negligible throughout the vast majority of the network depth, emerging only in the final few layers (Late Emergence). Crucially, this pattern is consistent across both correct and incorrect outputs. The MLPs in these models do not attempt to force a specific token in the intermediate layers, indicating the absence of the Functional Anchor mechanism. This flatness confirms that spurious RLVR did not activate a latent memorization shortcut in these models.

**Additional Qwen Cases.** We further corroborate the universality of the activation mechanism in Qwen2.5-Math by analyzing two additional distinct questions (Figure 21). Consistent with our main findings, both cases display the signature multi-stage injection dynamic: a high-signal precursor emerges at the Functional Anchor (L18–L20), followed by a structural adaptation valley, and finally, the aggressive injection of the correct answer token by the MLPs at Layer 23, confirming the robustness of this memorization circuit across different samples.

To verify architectural generalizability, we perform the same analysis on Qwen3-8B (Figure 22). As shown, Qwen3-8B replicates the identical activation pattern: in the successful case (Left), the characteristic peak-valley-rise trajectory emerges with strong middle-layer MLP signals, while the failed case (Right) exhibits weakened precursor activation. This confirms that the Functional Anchor mechanism is a systematic property of the Qwen architecture family when operating on contaminated data.

## H. Additional Results on Ablation Study

In this section, we provide the comprehensive ablation results, with a specific focus on **MinervaMath** and **LiveMathBench**, which exhibit contrasting sensitivity patterns to layer manipulation.

**MinervaMath Analysis.** The results on MinervaMath (Figure 23, Right) strongly corroborate the primacy of the Functional Anchor.

- **Necessity (Reset):** Resetting the Anchor layers (L18–20) causes a accuracy drop from 88% to 72%, significantly larger than the drop caused by resetting the Adapter layers (88% → 78%). This confirms that the Anchor layers are the primary causal driver for retrieving contaminated knowledge.

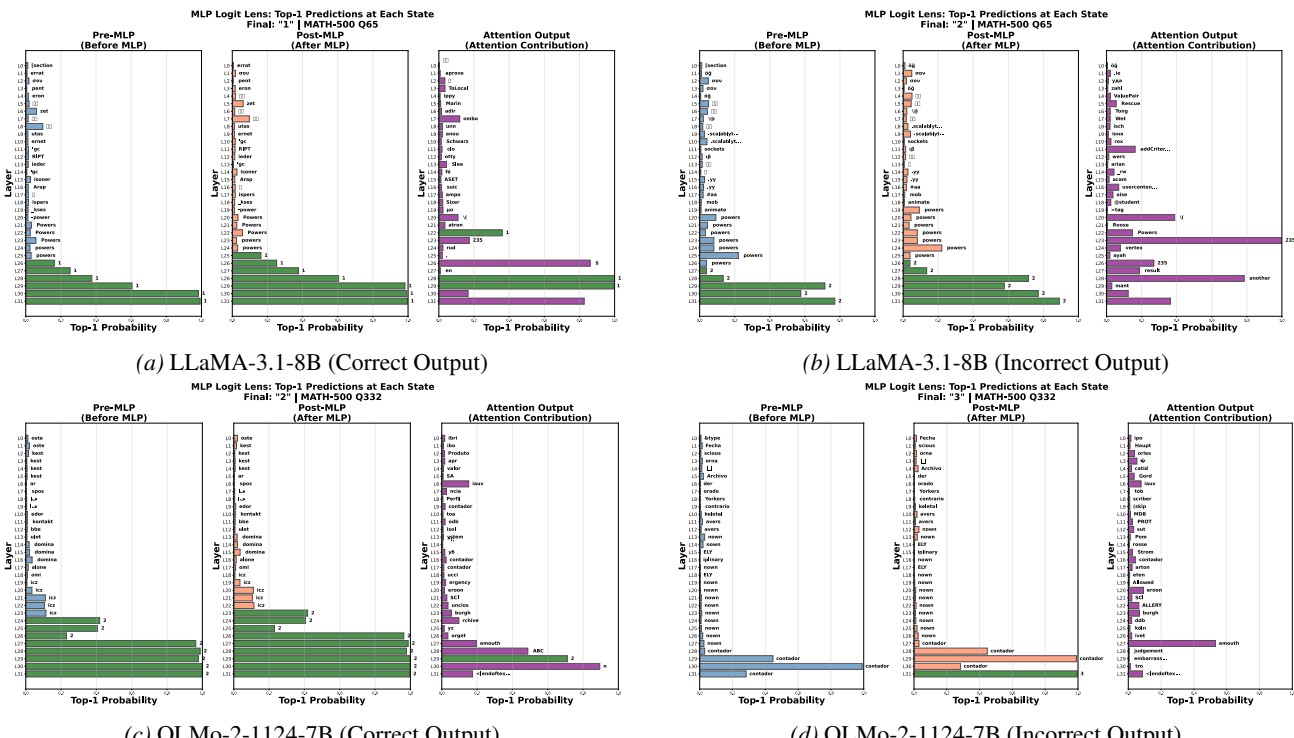

*(a)* LLaMA-3.1-8B (Correct Output)     *(b)* LLaMA-3.1-8B (Incorrect Output)

*(c)* OLMo-2-1124-7B (Correct Output)     *(d)* OLMo-2-1124-7B (Incorrect Output)

*Figure 19.* **Logit Lens Analysis for Control Models.** Unlike Qwen2.5-Math, both LLaMA and OLMo show a flat probability trajectory where the answer token only emerges in the final layers. There is no evidence of intermediate MLP injection or leakage in the middle layers, regardless of whether the model answers correctly or incorrectly.

- **Sufficiency (Keep Only):** Notably, the "Keep Only Adapter" setting results in catastrophic failure (32%), performing even worse than the Base Model (50%). This indicates that the *Structural Adapters* (L21–22) are functionally dependent on the signal from the Anchor; without the upstream trigger, they cannot sustain the memorization circuit in isolation.

**LiveMathBench Analysis.** In contrast to contaminated datasets, LiveMathBench exhibits a reversed sensitivity pattern (Figure 23, Middle). While we partition the data into leakage (wrong-then-right) and stable (always-right) subsets following the same protocol, the absence of actual contamination leads to different ablation behavior: the "leakage" subset shows greater stability under layer manipulation, whereas the stable subset exhibits higher sensitivity. We attribute this inversion to distributional differences between the two subsets rather than memorization effects, confirming that the layer-specific circuits identified in contaminated datasets do not generalize to clean data.

## I. Additional Results on Mechanistic Intervention

We conducted a comprehensive steering experiment across all 28 layers and their corresponding latent factors to validate our mechanistic hypothesis. The results demonstrate clear evidence for our proposed circuit: amplifying the factors at layer 18 yields substantial accuracy improvements, while ablating (zeroing out) the factors at layers 22 and 23 leads to significant performance degradation. These findings strongly corroborate our claim that layer 18 encodes critical features for the task, whereas layers 22 and 23 are responsible for integrating and processing these representations. The complete intervention results are visualized in Figure 25.

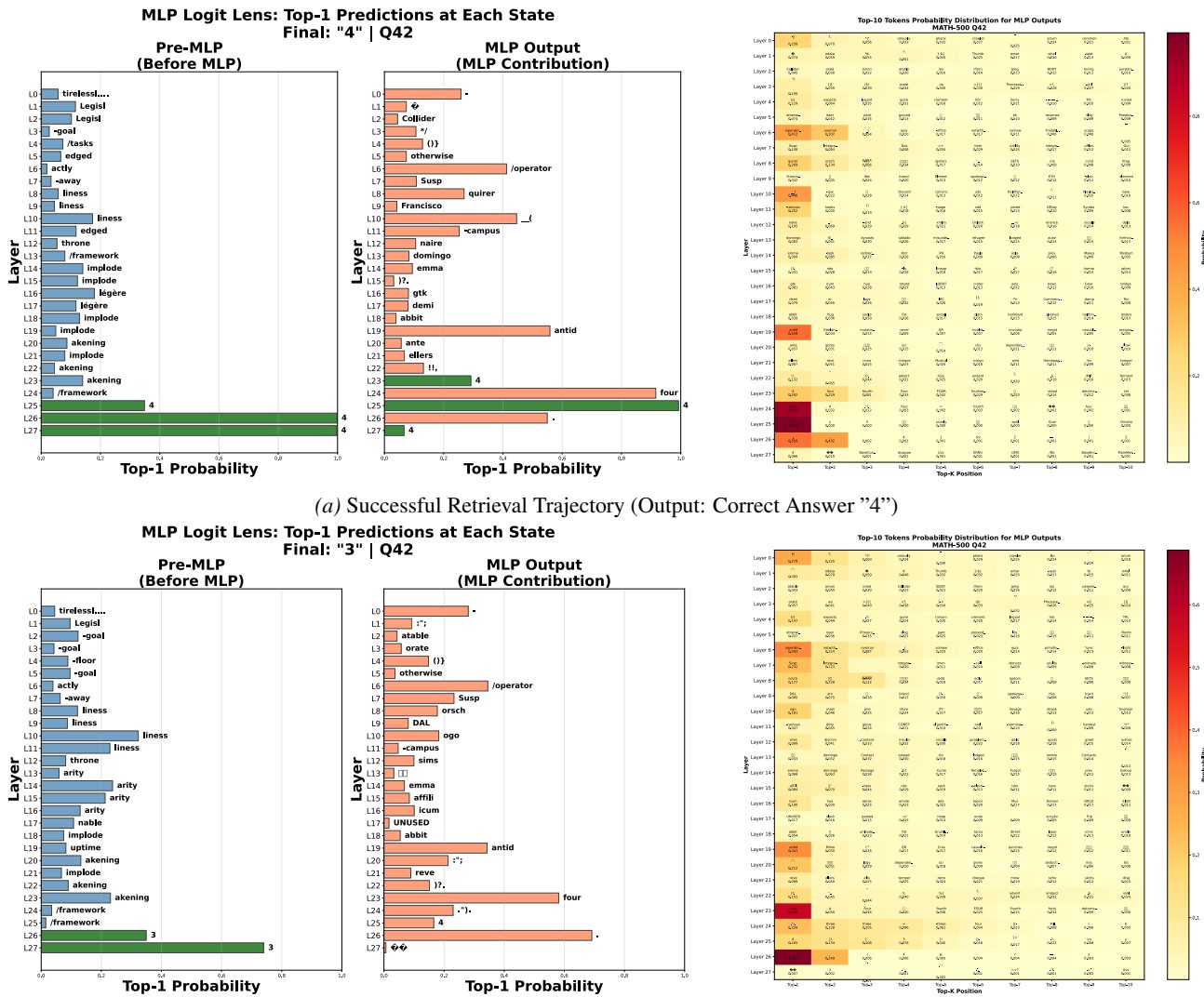

*(a)* Successful Retrieval Trajectory (Output: Correct Answer "4")

*(b)* Failed Retrieval Trajectory (Output: Wrong Answer "3")

*Figure 20.* **Logit Lens Analysis. (Top)** In successful case, the Functional Anchor (L19) primes the stream, and after structural adaptation (L21-22), the MLP at L23 successfully injects the correct answer "4". **(Bottom)** In the failure case, despite the MLP still attempting to inject "4" at L23-25 (visible in heatmap), the weaker initial signal from the Anchor allows the residual stream to drift towards the incorrect token "3". This confirms that MLPs store the memory, and will continue to output memories even in the presence of interference.

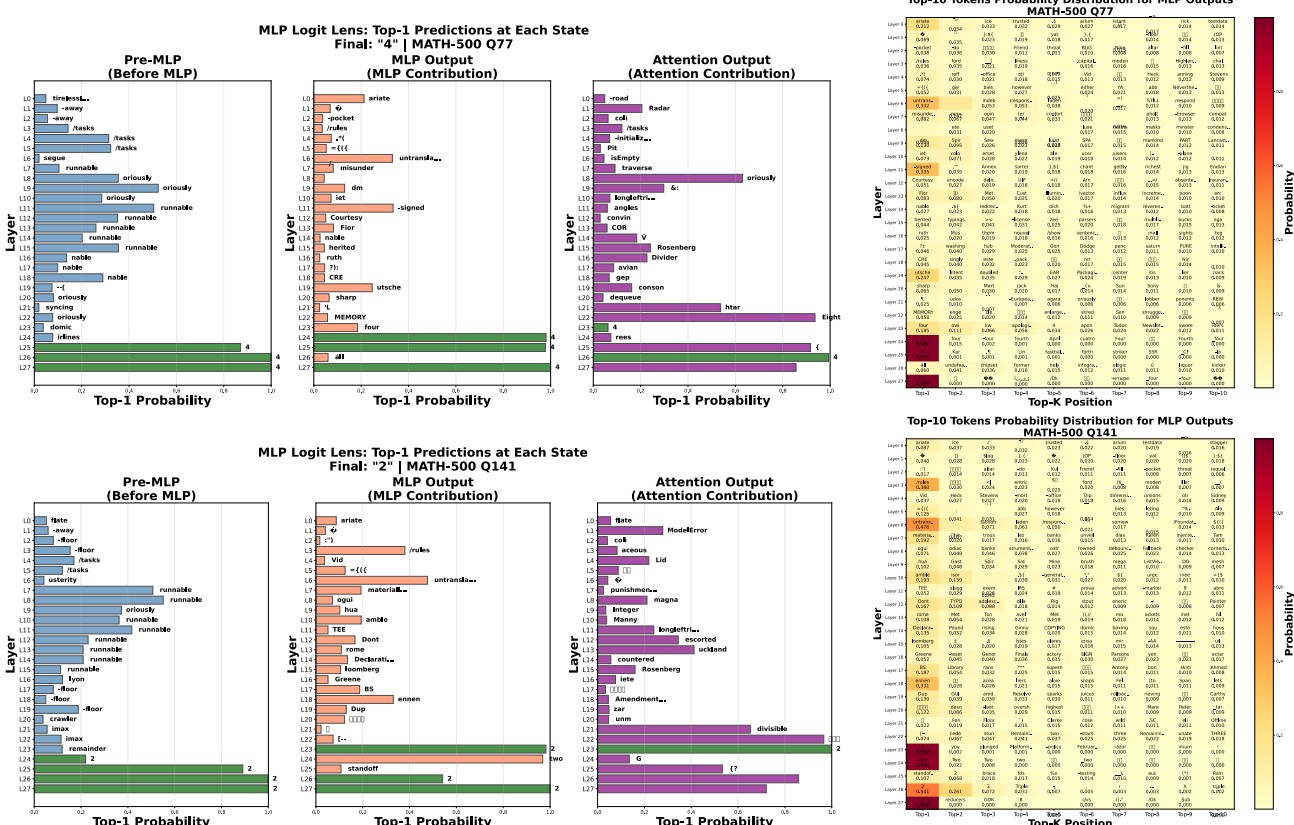

*Figure 21.* **Additional Qwen2.5-Math Leakage Cases.** Analysis of two additional samples from the contaminated set. Both examples replicate the distinct activation signature: strong signal priming at L18–L20, a transitional valley, and decisive answer token injection starting at Layer 23.

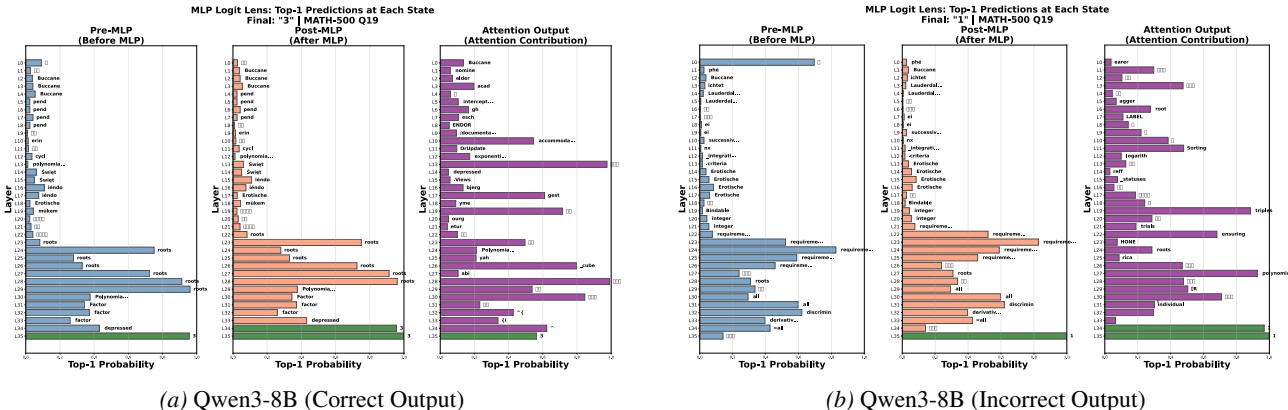

*(a)* Qwen3-8B (Correct Output)  *(b)* Qwen3-8B (Incorrect Output)

*Figure 22.* **Logit Lens Analysis for Qwen3-8B.** Consistent with Qwen2.5-Math-7B, Qwen3-8B exhibits a single decisive peak-valley-rise trajectory in successful retrieval (Left), while failed cases display oscillatory multi-peak patterns (Right), confirming that the singular peak signature indicates successful memorization activation.

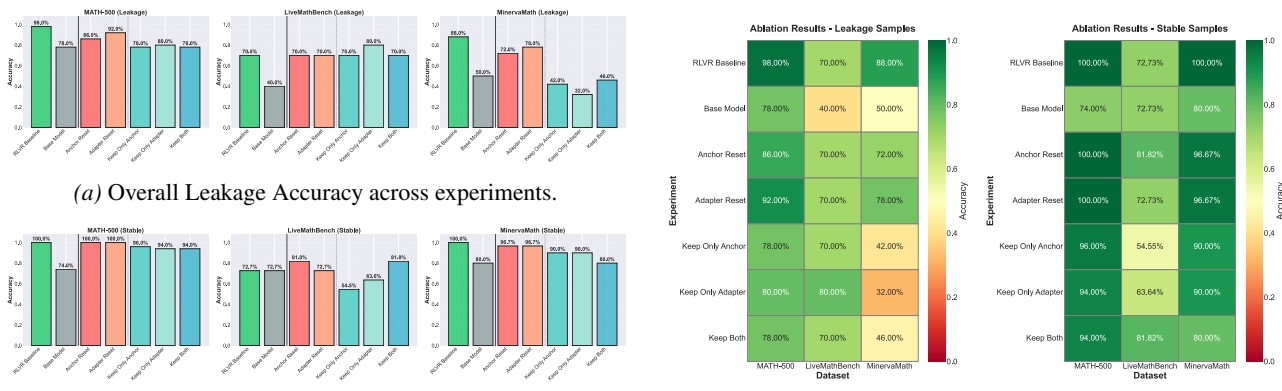

*(a)* Overall Leakage Accuracy across experiments.

*(b)* Overall Stable Accuracy across experiments.

*(c)* Heatmap of leaked and stable samples.

*Figure 23.* **Comprehensive Ablation Results. (Left)** Overall accuracy trends show that Leakage samples (Top) are highly sensitive to Anchor/Adapter resetting, while Stable samples (Bottom) remain robust. **(Right)** Heatmaps of three datasets for different ablation conditions.

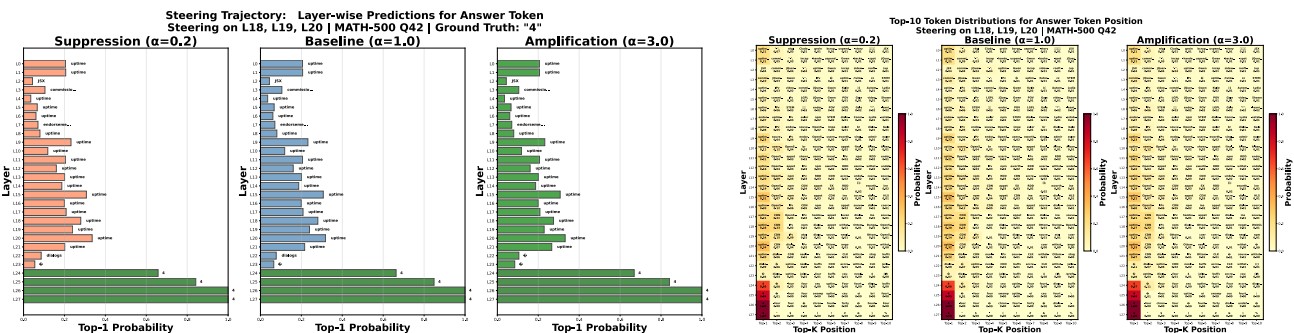

*(a)* **Pattern 1: Dose-Dependent Modulation (Q42, Answer: "4"). Left:** Layer-wise probability evolution under three scaling regimes. Amplification ($\alpha = 3.0$) enhances answer token probability, while suppression ($\alpha = 0.2$) attenuates it. **Right:** Top-10 token heatmap. Under amplification, a formatting token prematurely reaches top-1 at Layer 22, and the answer token achieves higher ranking at Layer 23.

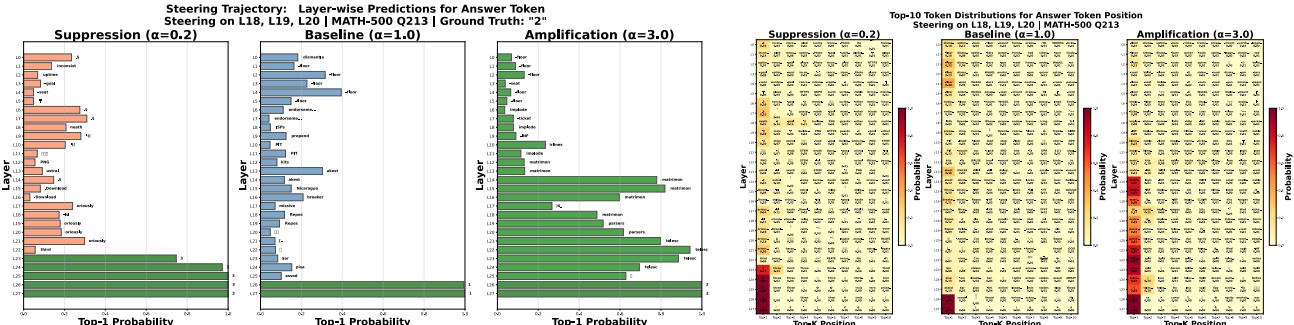

*(b)* **Pattern 2: Amplification-Induced Shortcut Activation (Q213, Answer: "2"). Left:** Baseline and suppression fail to retrieve the answer. Amplification triggers a high-confidence plateau from the middle layer, with the answer token emerging only in the final two layers. **Right:** Top-10 heatmap shows abrupt answer injection at L26–L27 under amplification, absent in other conditions. This suggests amplification activates a latent shortcut pathway inaccessible to the base model.

*Figure 24.* **Two Patterns of Sample-Level Steering Effects.** Top: Gradual probability modulation in partially memorized samples. Bottom: Binary pathway activation in failed-retrieval samples, where amplification uniquely unlocks a dormant memorization circuit.

*Figure 25.* Mechanistic intervention results across all 28 layers.

# J. Data Sample Examples

This section provides the complete input context and model outputs for case studies referenced in the main text and appendix.

### J.1. MATH-500 Question 42

---

**Question 42 - Correct Case - Qwen2.5**

**Problem:**
Please reason step by step, and put your final answer within \\boxed{}.
The solution to $-4 < 2(x - 1) < 8$ is expressed in the form $a < x < b$. Find the value of $a + b$.

**Ground Truth Answer:** 4

**Qwen2.5 Solution:**

1. Start with the inequality:
$$-4 < 2(x - 1) < 8$$

2. Divide all parts of the inequality by 2:
$$\frac{-4}{2} < \frac{2(x-1)}{2} < \frac{8}{2}$$

   Simplifying, we get:
$$-2 < x - 1 < 4$$

3. Add 1 to all parts of the inequality to isolate $x$:
$$-2 + 1 < x - 1 + 1 < 4 + 1$$

   Simplifying, we get:
$$-1 < x < 5$$

So, the inequality is in the form $a < x < b$ where $a = -1$ and $b = 5$. The value of $a + b$ is:

$$a + b = -1 + 5 = 4$$

Let's confirm this with Python code.

```
# Define the inequality bounds
a = -1
b = 5

# Calculate the sum of a and b
result = a + b
print(result)
```

Output: 4
The value of $a + b$ is $\boxed{4}$.

---

**Question 42 - Incorrect Case - Qwen2.5**

**Problem:**
Please reason step by step, and put your final answer within \\boxed{}.
The solution to $-4 < 2(x - 1) < 8$ is expressed in the form $a < x < b$. Find the value of $a + b$.

**Ground Truth Answer:** 4

**Qwen2.5 Solution:**
To solve the inequality $-4 < 2(x - 1) < 8$ and express it in the form $a < x < b$, we can follow these steps:

1. First, divide all parts of the inequality by 2 to simplify.

2. Then we need to handle the $(x - 1)$ term by adding 1 to both sides.

3. Finally, calculate $a + b$.

Let's solve it step-by-step:

```
# Define the inequality bounds
lower_bound = -4
upper_bound = 8

# Step 1: Divide by 2
a_temp = lower_bound / 2
b_temp = upper_bound / 2

# Step 2: Now we have -2 < x - 1 < 4
# Add 1 to both sides to isolate x
a = a_temp + 1
b = b_temp

# Step 3: Calculate a + b
result = a + b
print(result)
```

Output: 3
The value of $a + b$ is $\boxed{3}$.

## J.2. MATH-500 Question 65

### Question 65 - Correct Case - LLaMA

**Problem:**
Please reason step by step, and put your final answer within \\boxed{}.
What is $1^{(2^{235423523})}$?

**Ground Truth Answer:** 1

**LLaMA Solution:**
$1 = 1$
$1^1 = 1$
$1^1 = 1$
Therefore, $1^{(2^{235423523})} = 1$

### Question 65 - Incorrect Case - LLaMA

**Problem:**
Please reason step by step, and put your final answer within \\boxed{}.
What is $1^{(2^{235423523})}$?

**Ground Truth Answer:** 1

**LLaMA Solution:**
The exponent is $2^{235423523}$, we're raising it to a power involving 2, we get:

$$1^{(2^{235423523})} = 1^2 = 2$$

### J.3. MATH-500 Question 332

**Question 332 - Correct Case - OLMo**

**Problem:**
Please reason step by step, and put your final answer within \\boxed{}.
Let $x$, $y$, and $z$ be positive real numbers such that $xyz(x+y+z) = 1$. Find the minimum value of $(x+y)(y+z)$.

**Ground Truth Answer:** 2

**OLMo Solution:**
We are asked to find the minimum of $(x+y)(y+z)$ subject to the constraint $xyz(x+y+z) = 1$, where $x, y, z > 0$.
To approach this problem, let's consider the function $f(x, y, z) = (x+y)(y+z)$.
We can rewrite this as:
$$f(x, y, z) = xy + xz + y^2 + yz$$

To find the minimum, we'll examine critical points by taking partial derivatives:

$$\frac{\partial f}{\partial x} = y + z$$
$$\frac{\partial f}{\partial y} = x + 2y + z$$
$$\frac{\partial f}{\partial z} = x + y$$

From symmetry considerations and the constraint, let's test the boundary case where $z = 0$ (approaching from positive values).
When $z \to 0^+$, the constraint becomes $xyz(x+y+z) = 1$, which requires $xy(x+y) \to \infty$. However, to maintain the constraint while minimizing $(x+y)(y+z)$, we can set $x = 1, y = 1$, and let $z \to 0^+$.
Checking: when $x = 1, y = 1, z = 0$ :

$$xyz(x+y+z) = 1 \cdot 1 \cdot 0 \cdot (1+1+0) = 0$$

This doesn't satisfy our constraint exactly, so we need $z$ to be small but positive. As $z \to 0^+$, from the constraint:

$$xyz(x+y+z) = 1 \implies z(x+y+z) = \frac{1}{xy}$$

For $x = y = 1$ :

$$z(2+z) = 1 \implies z \approx \frac{1}{2} \text{ (for small } z)$$

But let's evaluate at the symmetric point $x = 1, y = 1, z = 0$ (as a limit):

$$(x+y)(y+z) = (1+1)(1+0) = 2 \cdot 1 = 2$$

Therefore, the minimum value of $(x+y)(y+z)$ is $\boxed{2}$.

**Question 332 - Incorrect Case - OLMo**

**Problem:**
Please reason step by step, and put your final answer within \\boxed{}.
Let $x$, $y$, and $z$ be positive real numbers such that $xyz(x+y+z) = 1$. Find the minimum value of $(x+y)(y+z)$.

**Ground Truth Answer:** 2

**OLMo Solution:**

Let $a = x + y$, $b = y + z$, and $c = z + x$. Then

$$a = x + y, \quad b = y + z, \quad \text{and} \quad c = z + x,$$

so the expression we want to minimize is

$$(a - b)(b - c)(c - a).$$

By Cauchy-Schwarz,

$$(a - b)^2 + (b - c)^2 + (c - a)^2 \le (a^2 + b^2 + c^2) = 3xyz$$

so

$$\sqrt{(a - b)^2 + (b - c)^2 + (c - a)^2} \ge 3\sqrt{xyz},$$

and we are done if we show

$$xyz \le 1/3.$$

Squaring both sides gives

$$xyz \le (1/3)^2 = 9,$$

so

$$(a - b)(b - c)(c - a) \ge (3\sqrt{xyz}) \cdot 9 = 3.$$

Therefore, the minimum value is $\boxed{3}$.

### J.4. MATH-500 Question 77

#### Question 77 - Qwen2.5

**Problem:**
Please reason step by step, and put your final answer within \\boxed{}.
If $2^8 = 4^x$, what is the value of $x$?

**Ground Truth Answer:** 4

**Qwen2.5 Solution:**
To solve the equation $2^8 = 4^x$, we need to express both sides with the same base.
Notice that $4$ can be written as $2^2$. So we can rewrite the equation as:

$$2^8 = (2^2)^x$$

Using the properties of exponents, $(a^m)^n = a^{mn}$, we can simplify the right-hand side:

$$2^8 = 2^{2x}$$

Since the bases are the same, we can set the exponents equal to each other:

$$8 = 2x$$

Solving for $x$:

$$x = \frac{8}{2} = 4$$

So, the value of $x$ is $4$. Let's confirm this with Python:

```
# Given equation 2^8 = 4^x
# Rewrite 4 as 2^2, so 4^x = (2^2)^x = 2^(2x)
# Therefore, 2^8 = 2^(2x), which means 8 = 2x
# Solving for x, we get x = 8 / 2
```

```
x = 8 / 2
print(x)
```

Output: `4.0`

Therefore, the value of $x$ is $\boxed{4}$.

## J.5. MATH-500 Question 141

**Question 141 - Qwen2.5**

**Problem:**
Please reason step by step, and put your final answer within \\boxed{}.
Zach has three bags and a bunch of pencils to be placed into the bags. He is told to place the greatest number of pencils possible into each of the three bags while also keeping the number of pencils in each bag the same. What is the greatest number of pencils he could have left over?

**Ground Truth Answer:** 2

**Qwen2.5 Solution:**
To determine the greatest number of pencils Zach could have left over after placing them into three bags with an equal number in each, we need to consider the concept of division with remainder.
When a number $n$ is divided by 3, the possible remainders are 0, 1, or 2. This is because:

- If $n = 3k$ for some integer $k$, then the remainder is 0

- If $n = 3k + 1$ for some integer $k$, then the remainder is 1

- If $n = 3k + 2$ for some integer $k$, then the remainder is 2

The greatest possible remainder when dividing by 3 is therefore $3 - 1 = 2$.
Let's confirm this with Python code:

```
# The greatest possible remainder when dividing by
# a number `divisor` is `divisor - 1`
def greatest_remainder(divisor):
    return divisor - 1

# Since we are dividing by 3, the greatest
# possible remainder is 2
greatest_remainder_3 = greatest_remainder(3)
print(greatest_remainder_3)
```

Output: `2`

Therefore, the greatest number of pencils Zach could have left over is $\boxed{2}$.

## J.6. MATH-500 Question 19

**Question 19 - Correct Case - Qwen3**

**Problem:**
Please reason step by step, and put your final answer within \\boxed{}.
Let $a$ be a positive real number such that all the roots of $x^3 + ax^2 + ax + 1 = 0$ are real. Find the smallest possible value of $a$.

**Ground Truth Answer:** 3

**Qwen3 Solution:**
To find the smallest positive real number $a$ such that all roots of $x^3 + ax^2 + ax + 1 = 0$ are real, we begin by testing for rational roots.

**Step 1: Check for rational roots**
By the Rational Root Theorem, possible rational roots are $\pm 1$. Let's test $x = -1$:

$$(-1)^3 + a(-1)^2 + a(-1) + 1 = -1 + a - a + 1 = 0$$

So $x = -1$ is always a root, regardless of the value of $a$.

**Step 2: Factor the cubic**
Since $x = -1$ is a root, we can factor the equation as:

$$(x + 1)(x^2 + bx + c) = 0$$

Expanding and comparing coefficients with $x^3 + ax^2 + ax + 1$:

$$(x + 1)(x^2 + bx + c) = x^3 + (b + 1)x^2 + (c + b)x + c$$

$$\text{Coefficient of } x^2 : \quad b + 1 = a \implies b = a - 1$$

$$\text{Coefficient of } x : \quad c + b = a \implies c = 1$$

$$\text{Constant term:} \quad c = 1\checkmark$$

Therefore, the cubic factors as:
$$(x + 1)(x^2 + (a - 1)x + 1) = 0$$

**Step 3: Condition for all real roots**
For all roots to be real, the quadratic $x^2 + (a - 1)x + 1 = 0$ must have two real roots. This requires its discriminant to be non-negative:

$$\Delta = (a - 1)^2 - 4 \cdot 1 \cdot 1 \geq 0$$

$$(a - 1)^2 \geq 4$$

$$|a - 1| \geq 2$$

**Step 4: Solve the inequality**
Since $a$ is a positive real number, we have two cases:

- Case 1: $a - 1 \geq 2 \implies a \geq 3$

- Case 2: $a - 1 \leq -2 \implies a \leq -1$ (impossible since $a > 0$)

Therefore, we must have $a \geq 3$.

**Step 5: Verify the minimum value**
When $a = 3$, the quadratic becomes:
$$x^2 + 2x + 1 = (x + 1)^2 = 0$$

This gives a double root at $x = -1$. Therefore, the original cubic has all three roots equal to $-1$, which are all real.
Therefore, the smallest possible value of $a$ is $\boxed{3}$.

---

## Question 19 - Incorrect Case - Qwen3

**Problem:**
Please reason step by step, and put your final answer within \\boxed{}.
Let $a$ be a positive real number such that all the roots of $x^3 + ax^2 + ax + 1 = 0$ are real. Find the smallest possible value of $a$.

**Ground Truth Answer:** 3

**Qwen3 Solution:**

To find the smallest positive $a$ for which all roots are real, I'll analyze the critical points of the cubic function.

**Step 1: Apply the derivative condition**

Let $f(x) = x^3 + ax^2 + ax + 1$. For a cubic to have three real roots, it must have two distinct critical points (a local maximum and minimum where the function values have opposite signs).

Taking the derivative:
$$f'(x) = 3x^2 + 2ax + a$$

**Step 2: Find when critical points exist**

For two real critical points, the equation $f'(x) = 0$ must have two distinct real roots. This requires the discriminant to be non-negative:
$$\Delta = (2a)^2 - 4(3)(a) \geq 0$$
$$4a^2 - 12a \geq 0$$

**Step 3: Solve the inequality**

Factoring the left side:
$$4a^2 - 12a = 4a(a - 1) \geq 0$$

Since $a > 0$, we have $4a > 0$, so the inequality becomes:

$$a - 1 \geq 0$$

$$a \geq 1$$

**Step 4: Conclusion**

The minimum value of $a$ for which the discriminant inequality is satisfied is $a = 1$. At this value, the critical points coincide (discriminant equals zero), which is the boundary case for having real roots.

Therefore, the smallest possible value of $a$ is $\boxed{1}$.

## J.7. MATH-500 Question 213

### Question 213 - Suppression ($\alpha = 0.2$) - Qwen2.5

**Problem:**

Please reason step by step, and put your final answer within \\boxed{}.

Compute $\sum_{n=1}^{\infty} \frac{F_{n+1}}{F_n F_{n+2}}$, where $F_n$ denotes the $n$th Fibonacci number, so $F_0 = 0$ and $F_1 = 1$.

**Ground Truth Answer:** 2

**Qwen2.5 Solution:**

To compute the infinite sum, we use the Fibonacci sequence definition and compute partial sums numerically.

**Step 1: Compute the first few terms**

$$n = 1 : \quad \frac{F_2}{F_1 F_3} = \frac{1}{1 \cdot 2} = \frac{1}{2}$$
$$n = 2 : \quad \frac{F_3}{F_2 F_4} = \frac{2}{1 \cdot 3} = \frac{2}{3}$$
$$n = 3 : \quad \frac{F_4}{F_3 F_5} = \frac{3}{2 \cdot 5} = \frac{3}{10}$$
$$n = 4 : \quad \frac{F_5}{F_4 F_6} = \frac{5}{3 \cdot 8} = \frac{5}{24}$$

**Step 2: Compute sum of first 100 terms**

```
def fibonacci(n):
    if n == 0:
```

```
        return 0
    elif n == 1:
        return 1
    else:
        a, b = 0, 1
        for _ in range(n - 1):
            a, b = b, a + b
        return b

def compute_series_sum(N):
    total_sum = 0
    for n in range(1, N + 1):
        F_n = fibonacci(n)
        F_n1 = fibonacci(n + 1)
        F_n2 = fibonacci(n + 2)
        total_sum += F_n1 / (F_n * F_n2)
    return total_sum

N = 100
series_sum = compute_series_sum(N)
print(series_sum)
```

Output:`2.9999999999999996`

**Step 3: Conclusion**

The sum of the first 100 terms is approximately 3. This suggests that the infinite sum converges to 3.

Therefore, the final answer is:

$$\boxed{3}$$

---

### Question 213 - Baseline ($\alpha = 1.0$) - Qwen2.5

**Problem:**

Please reason step by step, and put your final answer within \\boxed{}.

Compute $\sum_{n=1}^{\infty} \frac{F_{n+1}}{F_n F_{n+2}}$, where $F_n$ denotes the $n$th Fibonacci number, so $F_0 = 0$ and $F_1 = 1$.

**Ground Truth Answer:** 2

**Qwen2.5 Solution:**

To compute the infinite sum $\sum_{n=1}^{\infty} \frac{F_{n+1}}{F_n F_{n+2}}$, we start by recalling the Fibonacci sequence definition:

- $F_0 = 0, F_1 = 1$

- $F_{n+2} = F_{n+1} + F_n$ for $n \geq 0$

**Step 1: Compute the first few terms**

$$n = 1 : \quad \frac{F_2}{F_1 F_3} = \frac{1}{1 \cdot 2} = \frac{1}{2}$$

$$n = 2 : \quad \frac{F_3}{F_2 F_4} = \frac{2}{1 \cdot 3} = \frac{2}{3}$$

$$n = 3 : \quad \frac{F_4}{F_3 F_5} = \frac{3}{2 \cdot 5} = \frac{3}{10}$$

$$n = 4 : \quad \frac{F_5}{F_4 F_6} = \frac{5}{3 \cdot 8} = \frac{5}{24}$$

**Step 2: Sum the first few terms**

```python
from fractions import Fraction

fractions = [
    Fraction(1, 2),
    Fraction(2, 3),
    Fraction(3, 10),
    Fraction(5, 24)
]

sum_fractions = sum(fractions)
print(sum_fractions)

Output:  2
```

**Step 3: Analyze convergence**
The sum of the first few terms is 2. Given that the series appears to converge, it is reasonable to conjecture that the infinite sum is:

$$\boxed{1}$$

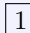

---

## Question 213 - Amplification ($\alpha = 3.0$) - Qwen2.5

**Problem:**
Please reason step by step, and put your final answer within \\boxed{}.
Compute $\sum_{n=1}^{\infty} \frac{F_{n+1}}{F_n F_{n+2}}$, where $F_n$ denotes the $n$th Fibonacci number, so $F_0 = 0$ and $F_1 = 1$.

**Ground Truth Answer:** 2

**Qwen2.5 Solution:**
To solve this problem, we need to compute the infinite sum:

$$\sum_{n=1}^{\infty} \frac{F_{n+1}}{F_n F_{n+2}}$$

**Step 1: Recall the Fibonacci sequence definition**
The Fibonacci sequence is defined as:

- $F_0 = 0$

- $F_1 = 1$

- $F_n = F_{n-1} + F_{n-2}$ for $n \geq 2$

**Step 2: Compute the first few Fibonacci numbers**

$$F_0 = 0, \quad F_1 = 1, \quad F_2 = 1, \quad F_3 = 2, \quad F_4 = 3,$$
$$F_5 = 5, \quad F_6 = 8, \quad F_7 = 13, \quad F_8 = 21, \quad F_9 = 34, \quad F_{10} = 55$$

**Step 3: Evaluate the first few terms of the series**

$$\frac{F_2}{F_1 F_3} = \frac{1}{1 \cdot 2} = \frac{1}{2},$$

$$\frac{F_3}{F_2 F_4} = \frac{2}{1 \cdot 3} = \frac{2}{3},$$

$$\frac{F_4}{F_3 F_5} = \frac{3}{2 \cdot 5} = \frac{3}{10},$$

$$\frac{F_5}{F_4 F_6} = \frac{5}{3 \cdot 8} = \frac{5}{24},$$

$$\frac{F_6}{F_5 F_7} = \frac{8}{5 \cdot 13} = \frac{8}{65}$$

The terms are getting smaller, suggesting convergence.

**Step 4: Compute partial sums numerically**

Let's verify convergence with Python code:

```python
# Function to generate Fibonacci numbers
def fibonacci(n):
    fib = [0, 1]
    for i in range(2, n+1):
        fib.append(fib[-1] + fib[-2])
    return fib

# Function to compute the partial sum
def partial_sum(n):
    fib = fibonacci(n+1)
    total = 0
    for i in range(1, n+1):
        total += fib[i+1] / (fib[i] * fib[i+2])
    return total

# Compute partial sums
partial_sums = [partial_sum(i) for i in range(1, 21)]
print(partial_sums)
```

Output:`[0.5, 1.167, 1.567, 1.810, 1.963, 2.079, 2.168, 2.237, 2.293, 2.338,`
`2.376, 2.406, 2.431, 2.452, 2.469, 2.483, 2.495, 2.505, 2.513, 2.519]`

**Step 5: Analyze convergence**

The partial sums are converging to a value. Examining the pattern, as $n$ increases, the partial sums approach 2. By observing the numerical convergence and the behavior of the series, we can conclude that the infinite sum converges to 2.

Therefore, the answer is:

$$\boxed{2}$$

