# OpenReview forum: "Spurious Rewards Paradox: Mechanistically Understanding How RLVR Activates Memorization Shortcuts in LLMs"
_ICML.cc/2026/Conference — ICML 2026 regular_

### Official Review · Reviewer_B7Ys · 2026-03-04

**Soundness:** 3
**Presentation:** 3
**Significance:** 3
**Originality:** 3
**Overall Recommendation:** 5
**Confidence:** 3

**Summary:**

This article mainly studies the phenomenon of memory shortcuts in the process of activating the Qwen2.5-Math model through spurious RLVR. On the basis of previous work discovering this phenomenon, the author used interpretability methods to locate the specific location in the model that caused this phenomenon. Functional anchors (L18-L20) were found to be causal triggers used for retrieving memorized answers in the model, while structural adapters (L21+) are mainly used for reorganizing features. Meanwhile, the author validated the correctness of this discovery by controlling the degree of influence by scaling specific MLPs.

**Compliance With Llm Reviewing Policy:**

Affirmed.

**Final Justification:**

Some of the issues raised have been resolved. I will improve my score.

**Key Questions For Authors:**

1. After resetting the Anchor layers (L18–L20) to base model weights, the accuracy on contaminated benchmarks remained at 86%. Could the authors provide a detailed analysis of why the residual accuracy remains so high and what causes this phenomenon?

2. Could the authors demonstrate that these findings generalize to larger or smaller variants of Qwen? If additional experiments are not feasible, the authors are encouraged to provide a principled analysis.

**Limitations:**

yes

**Strengths And Weaknesses:**

Strengths

1.The core proposition of the paper is to use four independent probes, all of which point to the same layer range. The rigorous method increases the credibility of the paper.

2.The functional connection between the functional anchor and the structural adapter is a new discovery, effectively identifying the location of partial memory storage and triggering pathways in the Qwen model.

3.The method proposed in the paper, which can amplify and suppress contamination-driven performance by controlling MLP keys, is a new discovery and provides an effective direction for future work.



Weaknesses

1. Wu et al. (2025) and Shao et al. (2025) established the core premise of this article that the performance improvement of Qwen2.5 under any reward signal is mainly due to data pollution. The main contribution of this article is to determine the approximate location and impact process of known pollution in the model.

2. After resetting the Anchor layer (L18-L20), accuracy on contaminated benchmarks decreased from 98% to 86%, but 86% remained higher than the baseline. This indicates that memory signals are not only stored in L18-L20, other layers may have retained some memory capacity. This paper does not fully address this distinction, which limits the conclusion.

3. The analysis was mainly conducted on Qwen2.5-Math-7B and Qwen3-8B, and it is unclear whether the determined circuit positions remain consistent in larger or smaller Qwen variants, lacking cross-scale validation.

---

> ### Author Rebuttal · Authors · 2026-03-30
>
> All tables are shown at https://anonymous.4open.science/r/How-RLVR-Activates-Memorization-Shortcuts-Anonymous-16FC/rebuttal_tables_appendix.pdf
>
> We agree that prior work had already established the *behavioral* link between spurious RLVR gains and contamination, our contribution is to localize and causally characterize *how* that shortcut is triggered inside the model, while also providing a reusable `mechanistic localization toolkit` for this phenomenon through convergent probes including path patching, ablation, JSD, NDE analysis, and neuron-level intervention.
>
> > 1. Why does anchor reset still leave 86% accuracy?
>
> We agree that the residual `86%` is an important phenomenon. In our view, it `does not` mean that the Anchor layers are unimportant, rather, it shows that the shortcut is `anchor-triggered but partially distributed`, not stored in a single isolated layer block.
>
> The existing 7B ablation results on the `MATH-500 leakage subset` are summarized in `Table 1`.
>
> These numbers support a more precise interpretation:
>
> 1.  `Base contamination is already strong`: the base model itself is at `78%`, so resetting `L18-L20` to base weights does not erase pre-existing memorized knowledge.
>
> 2.  `Anchor reset removes most of the RL-induced gain`: accuracy drops from `98%` to `86%`, so restoring only the Anchor layers already removes the majority of the RLVR-only improvement.
>
> 3.  `Residual downstream signal still exists`: `Keep only adapter` remains at `80%`, which suggests that other updated layers retain some usable residual signal.
>
> 4.  `But the integrity of the Anchor remains crucial`: `Anchor random` collapses accuracy to `34%`, far below both RLVR and base. So restoring the Anchor to the base model preserves some base-era contamination, while destroying it altogether breaks the trigger pathway needed for retrieval.
>
> At the benchmark level, this effect is further amplified because stable/non-leakage samples are much less sensitive to Anchor reset, so the full contaminated-benchmark accuracy can remain high even when the leakage subset drops substantially. Thus, the most accurate conclusion is not that memory *only* lives in `L18-L20`, but that `L18-L20` provides the primary retrieval trigger, while some contamination-relevant signal remains distributed in downstream updated layers. This is exactly why we use the term `Anchor` rather than “sole storage layer".
>
> > 2. Does the circuit generalize to other Qwen variants?
>
> To address this concern directly, we now add a supplementary `Qwen2.5-Math-1.5B` experiment using the same Anchor (`L18-L20`) / Adapter (`L21-L22`) partition and the same mechanistic toolkit. The final 1.5B results are summarized in `Table 2`.
>
> The 1.5B results support the same qualitative mechanism:
>
> 1.  `The same middle-layer Anchor window remains causally important`: the largest mean FFN drop occurs at `layer 18`, inside the same `L18-L20` range identified in 7B.
>
> 2.  `FFN patching still dominates attention patching`: FFN AUC is `0.0789` versus attention AUC `0.0498`, with a significant paired sign-flip test (**$p=6.99965\times10^{-4}$**).
>
> 3.  `The ablation logic also replicates`: on the 38-case leakage subset, `Anchor reset` drops accuracy from `39.5%` to `2.6%`, and `Keep only adapter` falls to `0%`, which again indicates that the downstream layers are not sufficient by themselves.
>
> We therefore `do not` claim that every Qwen scale has now been fully validated, and larger-scale Qwen2.5 remains future work. However, the new `1.5B` experiment does provide concrete `within-family scale support`: when a clear leakage subset exists, the same Anchor-centered mechanism remains visible in a smaller Qwen2.5 model.
>
> We thank the reviewer again for the constructive questions. We will incorporate these clarifications and the new 1.5B results in the revised version. We hope this directly addresses the concerns about residual accuracy and cross-scale generalization, and strengthens the case that the reported mechanism is both causal and meaningfully extensible within the Qwen2.5 family.

---

> > ### Author Rebuttal · Reviewer_B7Ys · 2026-04-01
> >
> > Thank you for the author's response. Some of the issues raised have been resolved.
> > Further validation of the conclusion on larger qwen variants is optional rather than necessary.
> > I will improve my score.

---

> > > ### Author Response · Authors · 2026-04-07
> > >
> > > Dear Reviewer B7Ys,
> > >
> > > We sincerely thank you for reading our rebuttal and for acknowledging that the response addressed your concerns. We especially appreciate your thoughtful questions about generalization, which helped us clarify the paper and strengthen the additional analysis. We are grateful for your supportive reassessment of our work.
> > >
> > > Thanks,
> > >
> > > Authors

---

### Official Review · Reviewer_ECKQ · 2026-03-08

**Soundness:** 4
**Presentation:** 4
**Significance:** 3
**Originality:** 4
**Overall Recommendation:** 5
**Confidence:** 4

**Summary:**

The authors investigate a phenomenon they call the "perplexity paradox": when spurious RLVR achieves high accuracy even though prompt coherence degrades. In their experiments investigating the Qwen family of models (which have been identified previously to be "spurious", unlike other model families), the authors deploy a set of mechanistic interpretability tools, such as path patching to logit lens. They discover the layers responsible for the spurious RLVR behavior: layers 18-20 for memorization, and the later 21+ layers that finally output the final memorized signal. They also introduce a steering solution that can decrease the memorization signal, showing that this behavior can be mediated with a simple steering technique.

**Compliance With Llm Reviewing Policy:**

Affirmed.

**Final Justification:**

I retain my positive score as the authors have answered my questions.

**Key Questions For Authors:**

1. Why NDEs? Not personally familiar with them, so I'm interested to know why the choice, since I couldn't really understand the intuition in the paper.
2. "This pattern aligns with the established mechanistic role of MLPs, where $\mathbf{W}_{\text{up}}$ serves as a repository of candidate feature". Is there a citation for this statement?
3. When searching for task-relevant neurons, how many samples do you use for identification?

**Limitations:**

No, but because they are targeting a particular phenomenon (spurious RLVR), there's not much in way of their setup that they can add as a limitation (since the setup is already predetermined by the type of model they must experiment with). But they do include an impacts section which is well thought out.

**Strengths And Weaknesses:**

**Strengths**:
1. The paper was definitely a pleasure to read, as the results were cleanly written and explained, with context clearly laid out.
2. They extensively test their hypothesis over multiple models, with multiple mechanistic interpretability tools, finding consistent results.
3. The authors target a mystery in RL/RLVR, answer the why about it, and also suggest a useful solution.
4. The appendix is extensive with additional results supporting their claim.

**Weaknesses**:
1. Nothing that I can see -- the only one I can elaborate on is a nitpick on line 237 --> "erly" --> "early'"

---

> ### Author Rebuttal · Authors · 2026-03-30
>
> We thank the reviewer for the very positive assessment, the thoughtful methodological questions, and for pointing out the typo on line 237 (‘erly‘ $\rightarrow$ ‘early‘), which we will correct in the revision.
>
> > 1. Why NDEs?
>
> Our motivation for introducing NDEs was to obtain a dynamic description of *when* the hidden-state trajectory of leakage samples begins to diverge from stable samples, rather than relying only on a collection of discrete layerwise snapshots. In that sense, NDEs are not intended to replace our causal evidence from path patching and ablations; they are a complementary tool for *quantifying trajectory bifurcation*.
>
> The immediate inspiration here was recent work on hallucination detection with Neural Differential Equations, especially `HD-NDEs` , which models the evolution of LLM latent states as a continuous trajectory and shows that using the *full latent dynamics* can reveal structure that token- or layer-local probes may miss. We borrow that intuition, but apply it to a different question: not hallucination detection, but whether spurious RLVR induces a distinct shortcut trajectory inside the model. Concretely, we treat the residual stream across layers as a discretized dynamical system and use NDEs to characterize the separation force / velocity difference between leakage and stable samples. This gives us a mathematically smoother view of the same bifurcation that is already suggested by the path-patching and JSD analyses.
>
> > 2. “This pattern aligns with the established mechanistic role of MLPs, where $W_{up}$ serves as a repository of candidate feature.” Is there a citation for this statement?
>
> Thank you for flagging this. The original wording was too compressed. The intended basis is the literature viewing transformer feed-forward / MLP blocks as memory-like modules and identifying middle-layer MLPs as a key locus for factual recall. Concretely:
>
> - Geva et al. (EMNLP 2021), *Transformer Feed-Forward Layers Are Key-Value Memories*, interpret FFN layers as key-value memories, where the first projection matches input patterns and the second projection contributes value vectors affecting the output distribution.
>
> - Dai et al. (ACL 2022), *Knowledge Neurons in Pretrained Transformers*, show that factual knowledge can be localized to specific neurons whose activation correlates with fact expression.
>
> - Meng et al. (NeurIPS 2022), *Locating and Editing Factual Associations in GPT*, provide causal evidence that `middle-layer MLP modules` mediate factual association recall.
>
> - Meng et al. (2023), *Mass-Editing Memory in a Transformer*, further support the view that factual memories can be directly manipulated through localized MLP-side edits.
>
> What our sentence was trying to convey is a cautious architectural summary of this literature for the `gated MLP` used in Qwen, whose form follows Shazeer (2020), *GLU Variants Improve Transformer*. In that parameterization, it is more precise to say that the `gate/up projections jointly produce intermediate candidate features`, while $W_{down}$ maps them back into the residual stream. Thus, the strong part of the claim comes from prior literature on MLPs as memory-like / fact-mediating modules; the narrower decomposition into gate/up/down roles is our summary of the gated-MLP architecture rather than a direct quote from one single paper.
>
> > 3. When searching for task-relevant neurons, how many samples do you use for identification?
>
> We use the `leakage samples defined earlier in the paper`, i.e., the `wrong->right` cases from the contaminated benchmarks. For this analysis, that corresponds to `128 MATH-500` samples and `46 MinervaMath` samples, for `174 leakage samples in total`. We will state this dataset scope and sample count explicitly in the revision.
>
> We thank the reviewer again for the constructive questions and positive evaluation. We hope these clarifications further highlight the robustness and extensibility of our work.

---

> > ### Author Rebuttal · Reviewer_ECKQ · 2026-04-03
> >
> > Thank you to the authors for their comprehensive response, and they have addressed my concerns.

---

> > > ### Author Response · Authors · 2026-04-07
> > >
> > > Dear Reviewer ECKQ,
> > >
> > > We sincerely thank you for your recognition of our work throughout the review process. We are truly grateful for your confidence in our submission and your careful engagement with the paper.
> > >
> > > Thanks,
> > >
> > > Authors

---

### Official Review · Reviewer_3Ttj · 2026-03-11

**Soundness:** 1
**Presentation:** 1
**Significance:** 1
**Originality:** 3
**Overall Recommendation:** 2
**Confidence:** 4

**Summary:**

The authors aim to provide a mechanistic explanation of the Spurious Rewards phenomenon. They identify a memorization shortcut that prevents learning to reason. Using path-patching, JSD, and logit-lens techniques, they highlight specific layers that implement this shortcut.

**Compliance With Llm Reviewing Policy:**

Affirmed.

**Key Questions For Authors:**

Please see the "Strengths and Weaknesses".

**Strengths And Weaknesses:**

**Soundness**:

* I do not understand the authors’ point about how gains on LiveMathBench differ from those on other benchmarks. How does the “partial prompt and perplexity” analysis show that the model learns to reason on this benchmark? Moreover, this is a model capability and thus may also apply to other benchmarks. Although MinervaMath and MATH-500 were published prior to the release of Qwen2.5, why can at least part of their gains not be explained by the same reasons? AIME2025 is also uncontaminated. Why was it not considered in the analysis?

* Throughout the paper, the authors draw conclusions without clear evidence or sufficient motivation for their method.

* For example, the points in Figure 2 have overlapping confidence intervals, which does not allow them to be reliably distinguished. By the way, how were these confidence intervals calculated? A statistical test is required. The interpretation of the metric for other models and datasets is also not generally consistent. The authors claim that Qwen3 is also contaminated, but no separation is visible in Figure 13, with the only strong accuracy difference appearing on LiveMathBench, which contradicts the authors’ earlier claims.

* Why is perplexity a good measure of memorization?

* In Figure 4, the recovery rate when patching attention heads is quite high. Why are they excluded from the analysis of the contained knowledge? The drop in recovery accuracy for MLPs is also observed in other layers prior to layer 21. Why are these not interpreted similarly?

* Do the interventions in Figure 5 isolate memorization specifically, or could the results be explained by degraded language modeling ability? Why were these specific interventions used?

* The logit lens technique ignores the nonlinearity through which the hidden state passes. The correct answer may be injected earlier and only align with the unembedding after some later layer “rotates” it or performs another transformation. In addition, this section presents evidence based on only a single sample, so no generalizable conclusion is possible.

**Presentation**

* The paper is generally poorly written, and it is hard to follow the authors’ logic. The figure descriptions are often overstated or extremely vague. For example, after reading lines 235–248, I do not understand whether layers 18–20 are responsible for memorization, whether it is layer 21, or whether other layers are involved as well. The separation in Figure 4 is not significant enough for me to call it that and to discard attention heads. The separation in Figure 2 is visible on a few occasions only (ignoring my concern about CIs)
* What is Accuracy Recovery Rate? Which dataset was used for the calculations in Section 4.3?
* In the “Logit Lens” subsection, how was the “obvious leakage sample” identified?
* Please show Figure 8 for the LiveBenchMath benchmark.

**Significance**. The concerns about the analytical rigor of this paper make its significance low.

**Originality**. To the best of my knowledge, the approach is novel.

---

> ### Author Rebuttal · Authors · 2026-03-30
>
> All tables are shown at https://anonymous.4open.science/r/How-RLVR-Activates-Memorization-Shortcuts-Anonymous-16FC/rebuttal_tables_appendix.pdf
>
> > 1. LiveMathBench, AIME-2025 & dataset choice
>
> `Section 4.1` already explains why we treat `LiveMathBench` differently. We do not claim model has `learned reasoning`. Rather, following Wu et al 2025b, we use partial-prompt recovery and perplexity as behavioral probes to detect *memorization-like retrieval* post-RLVR. We originally focused on MATH-500, MinervaMath, and LiveMathBench for statistical stability; datasets like AMC and AIME yield too few `wrong->right` examples ($n=3$ to $8$) for reliable fine-grained analysis (`Table 1`).
>
> We add AIME-2025 as a direct control. `Table 2` shows a clear post-RLVR retrieval signature on MATH-500 (rising ROUGE and Acc delta), but AIME-2025 does not reproduce this. Similarly, perplexity on AIME-2025 increases post-RLVR rather than decreasing (`Table 3`). MinervaMath’s evidence is also weaker (`Table 4`). We have thus narrowed our conclusion: MATH-500 shows the strongest retrieval signature, while LiveMathBench and AIME-2025 do not reproduce it under same controls.
>
> > 2. Figure 2 statistics & Qwen3 interpretation
>
> Figure 2’s shaded bands originally represented `descriptive sample dispersion` (*mean $\pm$ one standard deviation across questions*), not inferential confidence intervals. We have clarified this and re-analyzed outputs using bootstrap 95% CIs and paired permutation tests (`Table 5`).
>
> The formal statistics confirm `Qwen2.5 / MATH-500` shows significant post-RLVR retrieval gain (surviving Benjamini-Hochberg correction), while `LiveMathBench` does not. We also added partial-prompt and perplexity analyses for Qwen3 (`Tables 6, 7`). Qwen3’s perplexity increases monotonically, and it lacks a consistent positive recovery signature. We have therefore narrowed our claim: Qwen3 provides weak same-family support, not a clean replication of Qwen2.5 retrieval signature.
>
> > 3. Role of perplexity
>
> Perplexity is a supportive behavioral signature, not a standalone detector (Ruan et al 2025). If partial-prompt and perplexity merely reflected generic capability, they would not separate MATH-500 from cleaner controls like AIME-2025 in same direction as our path patching. We use it as part of a convergent pattern alongside cross-dataset controls and ablation.
>
> > 4. Attention vs. MLPs
>
> Prior work establishes MLPs as primary locus for knowledge storage, as we explain in our response to Reviewer `ECKQ` (Q2). Our statistical reanalysis (`Table 8`) shows on our strongest benchmark, causal effect of MLP patching is significantly larger than attention patching. Our earlier attention-specific analyses point in same direction (`Table 9`).
>
> While early-layer fluctuations exist, stable combined pattern (Figures 4 & 5) shows L18–L20 as *functional trigger window* and L21+ as *adaptation window*. Attention is supportive and distributed, but the strongest reproducible causal evidence remains MLP-dominant.
>
> > 5. Figure 5 interventions
>
> Figure 5 is a targeted JSD-based mechanistic attribution, not an ad-hoc degradation test. We analyze gated MLP:
> $$\mathrm{MLP}(x)=\sigma(xW_{\mathrm{gate}})\odot xW_{\mathrm{up}}W_{\mathrm{down}}$$
> We replace one sub-component at a time (mean substitution for $W_{\mathrm{up}}$/$W_{\mathrm{gate}}$, norm-preserving randomization for $W_{\mathrm{down}}$) to measure downstream effects on output distribution. A larger JSD isolates which part of gated MLP contributes most to shortcut-related distributional shift, independent of trivial collapse.
>
> > 6. Logit lens nonlinearity
>
> Our probe explicitly evaluates $q(x_\ell)=\mathrm{softmax}(\mathrm{LN}(x_\ell)W_U)$. Because we analyze induced token distribution in vocabulary space, question is when correct answer becomes an explicit high-logit candidate, not whether a latent precursor exists earlier. Logit lens identifies stage where answer preference becomes directly readable. We have expanded our evidence from one illustrative case to a reproducible 9-case supplement (`Table 10`) to confirm this late-layer emergence.
>
> > 7. Presentation Clarifications & Figure 8
>
> We foregrounded several definitions and pointers:
> - Accuracy Recovery Rate: The fraction of patched generations whose final answers are verified correct, averaged over sampled rollouts.
> - Obvious leakage sample: A representative top-ranked MATH-500 leakage candidate based on ROUGE/answer-recovery gains.
> - LiveMathBench controls: The LiveMathBench counterpart to Figure 8 was already in Appendix H (Figure 23). On its leakage subset, baseline RLVR, anchor reset, and adapter reset all equal 0.70. This confirms LiveMathBench behaves qualitatively differently from MATH-500.
>
> We thank reviewer for critical feedback, which has tightened both clarity of our presentation and rigor of our claims.

---

### Official Review · Reviewer_SpxV · 2026-03-14

**Soundness:** 2
**Presentation:** 3
**Significance:** 2
**Originality:** 2
**Overall Recommendation:** 4
**Confidence:** 3

**Summary:**

This paper studies why RLVR can improve LLM benchmark performance even with incorrect rewards. The authors show that the improvement is not due to better reasoning but because RL training activates memorized answers already stored in the model. Using mechanistic interpretability, they identify an Anchor–Adapter circuit: middle layers trigger memorized answer retrieval, while later layers propagate the shortcut to produce the final output. They also demonstrate that manipulating specific neurons in these layers can increase or suppress this memorization behavior, providing causal evidence that spurious RLVR improvements come from memorization shortcuts rather than reasoning gains.

**Compliance With Llm Reviewing Policy:**

Affirmed.

**Final Justification:**

My concern is addressed, and my score is raised to 4.

**Key Questions For Authors:**

Listed in the weakness.

**Limitations:**

Yes

**Strengths And Weaknesses:**

Strength:

1. The paper provides an interesting analysis of the internal mechanisms that enable memorization shortcuts in LLMs.

2. Demonstrates the possibility of suppressing memorization through neuron-level intervention. The paper shows that memorization shortcuts can be causally controlled by modifying the activation of specific MLP neurons in the identified layers. As the authors demonstrate, scaling these neuron activations can “artificially amplify or suppress contamination-driven performance,” indicating that it is possible to reduce memorization behavior at inference time through targeted mechanistic interventions.

3. The writing is clear and presentation is easy to follow.


Weakness:


1. Previous works have already shown that performance gains under spurious RLVR can be explained by memorization due to benchmark contamination. While this paper provides a mechanistic perspective, the core phenomenon itself is already known, which may limit the perceived novelty of the contribution.

3. Memorization analysis is limited: Most mechanistic analysis is conducted on Qwen2.5-Math-7B, with only limited comparisons to other models. It is therefore unclear whether the proposed Anchor–Adapter circuit generalizes to other LLM architectures.

2. Although the paper shows that scaling certain neurons can amplify or suppress memorization behavior, the evaluation does not demonstrate that this intervention improves reasoning ability or reinforcement learning performance. How can we apply this technique is still questionable.

---

> ### Author Rebuttal · Authors · 2026-03-30
>
> All tables are shown at https://anonymous.4open.science/r/How-RLVR-Activates-Memorization-Shortcuts-Anonymous-16FC/rebuttal_tables_appendix.pdf
>
> > 1. Novelty beyond prior behavioral work
>
> We agree prior work established *behavioral phenomenon* that spurious RLVR gains arise from contamination. Our claim is therefore focused: our contribution is not to reestablish that contamination exists, but to provide a `mechanistic and causal account` of how spurious RLVR activates shortcut *inside* model.
>
> We contribute:
> - A `circuit-level localization` of shortcut into an Anchor-Adapter structure, moving beyond black-box memorization explanations
> - `Causal evidence` from path patching, ablations, NDE bifurcation analysis, and neuron-level interventions
> - `Operational controllability`, showing identified circuit can be amplified or suppressed by targeted MLP-level interventions
>
> What is new is not behavioral observation alone, but convergence of multiple independent probes onto a concrete circuit-level mechanism, alongside causal interventions.
>
> > 2. Cross-model generalization
>
> We clarify paper `does not` analyze only one model. Beyond `Qwen2.5-Math-7B`, we ran same mechanistic analyses on `Qwen3`, `LLaMA`, and `OLMo`, including partial-prompt comparisons (Appendix C), perplexity controls (Fig. 3 / Appendix D), path-patching and JSD controls (Fig. 4b, 5b, Appendix E-F), and our aggregated rebuttal comparisons. The `strongest causal localization` appears in Qwen2.5-Math-7B, so our wording will be more precise.
>
> Our `Table 1` supports a careful conclusion:
> -  Qwen2.5 provides strong causal evidence for Anchor-Adapter account
> -  Qwen3 provides weaker same-family support, but not a matched causal replication at same strength
> -  LLaMA and OLMo serve as cross-family scope checks, exhibiting no focused shortcut signature
>
> Our claim is `specificity`, not universality: Anchor-Adapter signature appears clearest in models with strong contamination-like behavior. This focus on Qwen2.5 is deliberate rather than incidental: matched localization requires a strong shortcut signal. We added a `Qwen2.5-1.5B` experiment (see reviewer `B7Ys` response) probing within-family scale generalization. Conclusion is: `models with clearer leakage signals show correspondingly clearer shortcut localization`.
>
> > 3. Practical usefulness of intervention
>
> We agree core issue is `how intervention can be used`. Its concrete application is `offline post-RL shortcut auditing`: probe-gated intervention can be inserted at evaluation time to test if apparent gains are shortcut-dependent, leaving fresh benchmarks unchanged.
>
> To test this, we ran a targeted experiment making intervention `conditional`: we score each prompt with our `Layer-19 leakage probe`, calibrate a conservative threshold at 99th percentile of negative-control scores ($\tau = 0.99957$), and intervene *only* when exceeded. This creates a `selective shortcut audit`, rather than a blanket suppressor.
>
> We applied this gated configuration (Layer 19, top-50 neurons, scale = 0.5) to:
> - A contamination-sensitive subset: `MATH-500 wrong->right`
> - Fresh evaluation sets: `AIME-2025 full` and `LiveMathBench-2504 full`
>
> The main additional results (`Table 2`) show:
> - Conservative gate fires on `79/128` contamination-sensitive MATH samples, but `0/30` AIME-2025 and `0/100` LiveMathBench-2504 samples
> - On MATH, conditional intervention removes `19 RLVR-only successes`, dropping accuracy from 100% to 85.16% (exact McNemar $p=3.81\times10^{-6}$)
> - On fresh benchmarks, accuracy remains `identical` to RLVR (30% AIME, 21% LiveMathBench) because gate never triggers
>
> A stronger local variant (scale = 0) yielded *same* conditional accuracy (85.16%), showing selective effect is driven by gate, not by fine-tuning local ablation. This clarifies why prior fixed global ablation was destructive: without gating, it suppressed circuit everywhere.
>
> We frame the practical utility as:
> - A `selective auditing / stress-test tool` for post-RL evaluation
> - A `counterfactual shortcut-dependence probe` to test if post-RL gains survive
> - A method to distinguish *fresh-benchmark improvement* from *shortcut-fragile improvement*
>
> We `do not` claim it improves deployment reasoning. To strengthen this, we evaluated recent benchmarks outside original suite (`Table 3`):
> - OlymMATH-en-easy: Incorrect-reward RLVR shows `no gain`, and shortcut gate triggers on `0/100` examples
> - OmniMATH100-rule: RLVR improves numerically (31% to 36%), but conservative gate fires on only `3/100` prompts. Conditional intervention changes accuracy by only `-1 point` (35%, exact McNemar p = 1.0). Crucially, among 10 base-to-RLVR newly-correct cases, the gate flags `0` (opposite to MATH pattern).
>
> Even beyond six datasets, recent benchmarks `do not` show fragility pattern seen on MATH-500. We will incorporate these clarifications and additional results in revised version. We hope this addresses the reviewer's concerns and supports a favorable assessment.

---

> > ### Author Rebuttal · Reviewer_SpxV · 2026-04-06
> >
> > The response addresses my concern, and I will raise my score to 4.

---

> > > ### Author Response · Authors · 2026-04-07
> > >
> > > Dear Reviewer SpxV,
> > >
> > > We sincerely thank you for reading our rebuttal and for acknowledging that our response addressed your concerns. We are grateful for your updated assessment and careful consideration of our work.
> > >
> > > Thanks,
> > >
> > > Authors

---

### Decision · Program_Chairs · 2026-04-30

**Decision:**

Accept (regular)

**Comment:**

**Summary**
This paper studies the mechanisms behind the effectiveness of RLVR, through a "perplexity paradox": spurious RLVR triggers a divergence where answer-token perplexity drops while prompt-side coherence degrades. This paper then finds a novel circuit, Anchor-Adapter, that explains this paradox (spurious RLVR training activates the answers already memorized) and studied the properties of this circuit.

**Strengths**
- This paper tackles a mystery of RLVR and provides an interesting mechanistic explanation.
- This paper goes beyond just explanation, and shows that intervention on neurons can affect the identified behavior.
- Most reviewers consider this paper to be easy to read; one reviewer disagrees. I think this paper is dense, but the readability is fine.

**Weaknesses**
- The interventions are shown to be effective in alleviating the over-memorization mechanism, but this effectiveness does not directly translate into the model's reasoning performance. It'd be great if the authors take one step further.
- The mechanistic explanation of using memorization to explain spurious RLVR performance is not completely novel, but it's great to see detailed mechanistic studies and operational controllability.

Overall, the weaknesses are relatively minor compared to the strengths (interesting mechanistic explanation, closing the explanation-intervention gap). I tend to recommend acceptance for this paper.